

# Anisotropic Landau-Lifshitz model in discrete space-time

Žiga Krajnik[1] ⋆, Enej Ilievski[1], Tomaž Prosen[1] and Vincent Pasquier[2]

**1** Faculty of Mathematics and Physics, University of Ljubljana,
Jadranska 19, 1000 Ljubljana, Slovenia
**2** Institut de Physique Théorique, Université Paris Saclay, CEA,
CNRS UMR 3681, 91191 Gif-sur-Yvette, France

⋆ ziga.krajnik@fmf.uni-lj.si

## Abstract

We construct an integrable lattice model of classical interacting spins in discrete space-time, representing a discrete-time analogue of the lattice Landau-Lifshitz ferromagnet with uniaxial anisotropy. As an application we use this explicit discrete symplectic integration scheme to compute the spin Drude weight and diffusion constant as functions of anisotropy and chemical potential. We demonstrate qualitatively different behavior in the easy-axis and the easy-plane regimes in the non-magnetized sector. Upon approaching the isotropic point we also find an algebraic divergence of the diffusion constant, signaling a crossover to spin superdiffusion.

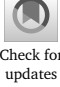
---

# 1  Introduction

The advent of powerful computational tools [1, 2] has tremendously advanced our understanding of many-body physics in and out of equilibrium [3–6]. Dealing with thermodynamic systems of strongly interacting degrees of freedom nonetheless still presents a dauting task in general. In order to efficiently simulate many-body quantum dynamics on a computer one typically confronts the issue of growing complexity which arises when distant regions of space become entangled. While *classical* many-body dynamical systems on the other hand do not suffer this issue, computing time-evolution of statistical ensembles often proves to be similarly challenging and require an immense amount of computational resources.

Amidst recent experimental breakthroughs with cold-atom technologies [7, 8], there has been an upsurge of interest in the study of out-of-equilibrium phenomena. Nowadays, optical lattices not only enable manufacturing of tailor-made nanodevices for processing quantum information, but also offer a versatile platform to directly probe the relaxation process at finite energy density in the systems both near and far from equilibrium [9–12].

Concurrently, we have seen an increased theoretical interest in various facets of nonequilibrium physcis. Integrable systems have been in the spotlight lately [13–16, 16–22], largely due to their inherent non-ergodic features [23–26] and anomalous transport properties [5, 27–37] as recently covered in a compilation of review articles [38–43]. The study of nonequilibrium properties in classical integrable dynamical systems of interacting particles or fields has received comparatively less attention [44–51].

One major point of worry concerning numerical simulations of *integrable* dynamics is that integration schemes based on naïve (e.g. Trotter–Suzuki, or Runge–Kutta) discretizations invariably destroy integrability. On sufficiently short time scales this should not be a major concern. By contrast, even weakly broken integrability is likely to induce spurious effects at late times and thus preclude reliable extraction of transport coefficients or dynamical exponents. In addition statistical field theories are commonly plagued by UV divergences (even when the local target space is a compact manifold). These drawbacks can both be obviated in a fully discrete setting.

Integrable many-body dynamical systems in discrete time [52, 53] and classical cellular automata [42, 54] have recently attracted much interest. An integrable space-time discretization of the isotropic classical Heisenberg ferromagnet has been recently obtained in Ref. [55], and subsequently generalized to a large class of 'matrix models' [53] which are globally invariant under the action of simple Lie groups. In this work, we construct an *anisotropic* deformation of the classical $SU(2)$ ferromagnet obtained in Ref. [55], representing a 'brick-wall type' circuit composed from elementary two-body symplectic maps. Our model can be thus regarded as an 'integrable Trotterization' [53, 55–57] of the anisotropic lattice Landau-Lifshitz field theory [58–60] (the classical counterpart of the celebrated quantum XXZ spin chain). A fully discrete integrable Landau–Lifshitz equation has, to the best of our knowledge, not been constructed yet; it has only appeared previously in a paper by Hirota [61] in a bilinear form in terms of the discrete *tau* function. The outcome of our construction is an explicit finite-

step integrable integration scheme that can be efficiently employed to computationally study dynamical properties of the model. As an application, we compute the Drude weight and diffusion constant as functions of spin chemical potential (i.e. magnetization) and anisotropy.

**Outline.** This paper is structured as follows. We begin in Section 2 by describing the formal setting. First, we introduce a discrete zero-curvature property of an auxiliary linear transport problem on a (tilted, light-cone) space-time lattice and interpret it in terms of a local dynamical map acting on a pair of classical spins. We proceed by introducing a dynamical system on a discrete space-time lattice in the form of a 'symplectic circuit'. Next, in Section 2.1 we define the local phase space equipped with a Poisson structure, and introduce the Lax matrix of the lattice Landau–Lifshitz model. Moving on, in Section 2.2 we outline how to explicitly solve the zero-curvature relation by exploiting the underlying algebraic relations, yielding the local time-propagator of the model. Section 3 is devoted to a physics application, where we carry out a detailed numerical study of magnetization transport in grand-canonical Gibbs states for the easy-axis and easy-plane regimes of the model. In Section 4 we conclude with a brief summary of the main results.

# 2 Anisotropic symplectic spin model in discrete space-time

Before delving into the specifics of the model, we first introduce the general setting. We shall mostly follow the lines and use the notation from previous works on related discrete-time models with isotropic interactions [53, 55].

**Discrete space-time.** The physical space-time lattice is a two-dimensional square lattice with nodes $(\ell, t) \in \mathbb{Z}^2$, with $\ell$ and $t$ referring to the spatial and temporal indices, respectively. Throughout the paper we adopt the convention that time flows upwards while the spatial axis is oriented towards the right. Moreover, we adopt periodic boundary conditions in space by identifying $\ell \equiv \ell + L$ and additionally assume, for definiteness, the system length $L$ to be even.

Each site of the space-time lattice is attached a local physical degree of freedom $\mathbf{S}_\ell^t$, which we consider to be a classical spin $\mathbf{S} = (S^1, S^2, S^3)$ of unit length, $\mathbf{S} \cdot \mathbf{S} = S^+ S^- + (S^3)^2 = 1$ (with $S^\pm \equiv S^1 \pm iS^2$) obeying the canonical ultralocal Lie–Poisson brackets

$$\{S_\ell^a, S_{\ell'}^b\} = \delta_{\ell, \ell'} \sum_c \epsilon_{abc} S_\ell^c, \tag{1}$$

where we have used the Levi–Civita symbol $\epsilon_{abc}$.

We furthermore introduce a square light-cone lattice by tilting the space-time lattice by 45° degrees, and assign to its vertices (nodes) $(n, m) \in \mathbb{Z}^2$ auxiliary variables $\phi_{n,m}$. Physical spins accordingly sit at the middle of the edges on the light-cone lattice.

**Discrete zero-curvature condition.** Integrabiliy of a dynamical systems in discrete time is a direct manifestation of a discrete zero-curvature property of an associated linear transport problem [62, 63] for auxiliary variables $\phi$, in analogy to completely integrable Hamiltonian systems (see e.g. Refs. [60, 64, 65]). To realize parallel transport on the auxiliary light-cone lattice we introduce the lattice shift operators [66] along the light-cone directions (i.e. characteristics $\ell \pm t = \text{const}$)

$$\phi_{n+1,m} = L_{n,m}^{(+)}(\lambda)\phi_{n,m}, \qquad \phi_{n,m+1} = L_{n,m}^{(-)}(\lambda)\phi_{n,m}. \tag{2}$$

Here the local 'matrix propagators' $L^{(\pm)}$ represent certain matrix-valued functionals of physical variables which, in addition, depend analytically on a free complex parameter $\lambda$. In order to

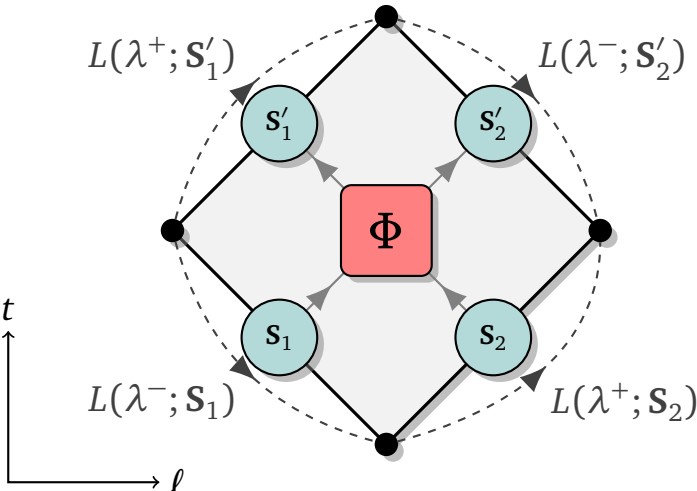

Figure 1: Elementary plaquette of the discrete light-cone lattice: classical spins $\mathbf{S} \in S^2$ (circles at the middle of the edges) are situated on the vertices of the discrete space-time lattice. Primed variables $\mathbf{S}'_{1,2}$ pertain to spins $\mathbf{S}_{1,2}$ that are time-shifted forward by one unit of time by applying the propagator $\Phi$ (red square).

ensure consistency of such parallel transport, there should be no ambiguity in the order of propagation when passing from $\phi_{n,m}$ to $\phi_{n+1,m+1}$. In this case, the shift matrices $L^{(\pm)}$ constitute the *Lax pair* and $\lambda$ is called the spectral parameter. More generally, one can also allow the spectral parameter to have additional dependence on local 'inhomogeneities' along an initial sawtooth. We shall make a special 'homogeneous choice' by requesting $L^{\pm} \equiv L(\lambda^{\pm}; \mathbf{S})$, i.e. using a pair of spectral parameters $\lambda^{\pm}$ depending only on the light-cone direction (but not on position $\ell$ or time $t$).

Commutativity of the light-cone shifts is neatly encapsulated by the following *discrete zero-curvature property* [67, 68] around an elementary square plaquette of the light-cone lattice,[1]

$$L(\lambda^{+}; \mathbf{S}_2)L(\lambda^{-}; \mathbf{S}_1) = L(\lambda^{-}; \mathbf{S}'_2)L(\lambda^{+}; \mathbf{S}'_1), \tag{3}$$

where we have assumed, for notational convenience, that the 'input' variables $\mathbf{S}_{1,2}$ sit at lattice sites 1 and 2. Notice that each of the light-cone Lax matrices $L(\lambda^{\pm})$ depend on the local 'edge variable' $\mathbf{S}$ on the corresponding edge, whereas primed variables $\mathbf{S}'$ are being used as a shorthand notation for the propagated spins, i.e. original variables $\mathbf{S}$ shifted by one unit in the time direction (as depicted in Figure 1). The discrete curvature is fulfiled *if and only if* the time-updated 'output' variables $\mathbf{S}'_{1,2}$ are appropriately linked to 'input' variables $\mathbf{S}_{1,2}$. This requirement allows us to interpret the flatness condition as an (implicit) specification of a local time-propagator, i.e. a local two-body map $\Phi : (\mathbf{S}_1, \mathbf{S}_2) \mapsto (\mathbf{S}'_1, \mathbf{S}'_2)$ for any spatially adjacent pair of sites.

The zero-curvature condition implies the integrability of the model through a property called *isospectrality*. As explained in Appendix C, isospectrality gives rise to an infinite set of local conserved quantities in involution.

**Many-body propagator.** Let us for the time being leave the two-body propagator $\Phi$ defined implicitly via the zero-curvature property. In the following, we first outline how to use $\Phi$ as the elementary building block of a many-body dynamical map in the form of a 'brick-wall circuit'.

---

[1] A magnetic field could be incorporated in the construction by adding constant 'twist' matrices to the zero-curvature condition. See [53] for a similar construction.

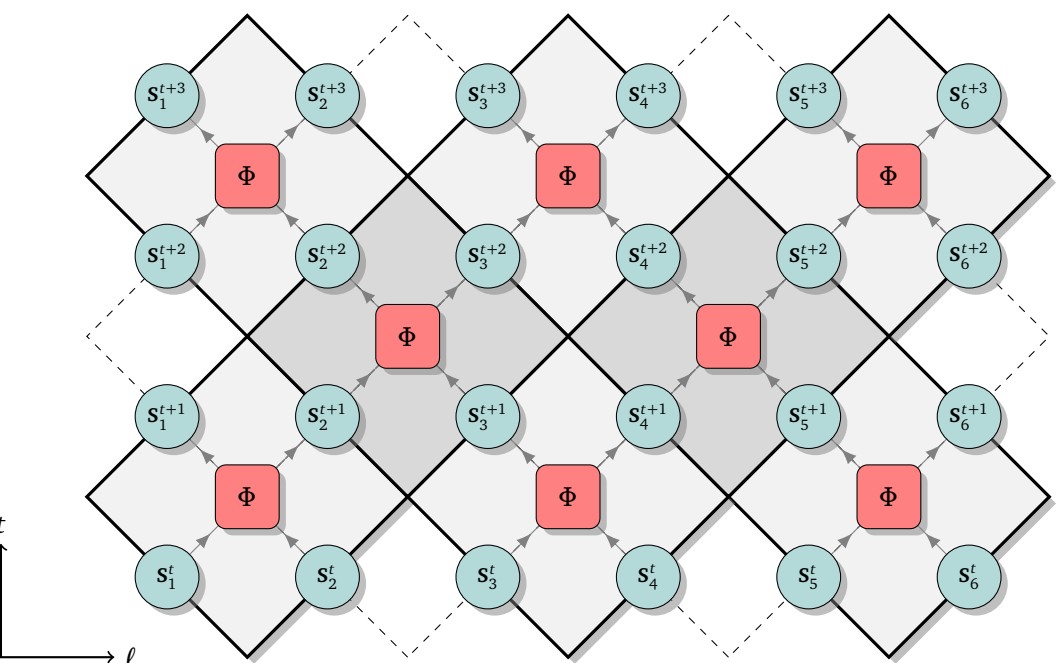

Figure 2: Fabric of discrete space-time: the square space-time lattice comprising classical spin degrees of freedom $\mathbf{S}_\ell^t$ (circles) superimposed on a square light-cone lattice represented by a tilted checkerboard. In the middle of each plaquette, there is a two-body symplectic map $\Phi$ (red square) which maps the pairs of physical spin variables forward in time.

To this end, let $\mathcal{M}_1 \cong S^2$ denote a local phase space (a 2-sphere) of a canonical spin. The full phase space $\mathcal{M}_L$ of an $L$-site chain is thus given simply by the $L$-fold Cartesian product, $\mathcal{M}_L \equiv \mathcal{M}_1^{\times L}$, and accordingly we introduce a many-body dynamical map $\Phi^{\text{full}} : \mathcal{M}_L \mapsto \mathcal{M}_L$.

By virtue of the light-cone geometry, the full time-dynamics consists of an alterating sequence of 'odd' and 'even' local propagators,

$$(\mathbf{S}_{2\ell-1}^{2t+2}, \mathbf{S}_{2\ell}^{2t+2}) = \Phi(\mathbf{S}_{2\ell-1}^{2t+1}, \mathbf{S}_{2\ell}^{2t+1}), \qquad (\mathbf{S}_{2\ell}^{2t+1}, \mathbf{S}_{2\ell+1}^{2t+1}) = \Phi(\mathbf{S}_{2\ell}^{2t}, \mathbf{S}_{2\ell+1}^{2t}), \tag{4}$$

respectively. This results in a staggered circuit as depicted in Figure 2. By making use of the embedding prescription,

$$\Phi^{(j)} = I^{\times(j-1)} \times \Phi \times I^{\times(L-j-1)}, \tag{5}$$

for $j = 1, \ldots, L-1$, where $I : \mathcal{M}_1 \mapsto \mathcal{M}_1$ stands for a local (single-site) unit map $I(\mathbf{S}) \equiv \mathbf{S}$, ($\Phi^{(L)}$ correspondingly affects the $L$th and the first spin), the full propagator $\Phi^{\text{full}}$ for two units of time $t \mapsto t+2$ decomposes as

$$\Phi^{\text{full}} = \Phi^{\text{even}} \circ \Phi^{\text{odd}}, \tag{6}$$

with odd and even propagators further factorizing into commuting two-body maps,

$$\Phi^{\text{odd}} = \prod_{\ell=1}^{L/2} \Phi^{(2\ell)}, \qquad \Phi^{\text{even}} = \prod_{\ell=1}^{L/2} \Phi^{(2\ell-1)}. \tag{7}$$

So far our construction has been entirely formal. In what follows, we derive the map $\Phi^{\text{full}}$ that corresponds to an integrable space-time discretization of the anisotropic lattice Landau–Lifshitz model, as introduced by Sklyanin [59, 60, 69]. In this view, $\Phi^{\text{full}}$ represents its 'integrable Trotterization'.

## 2.1 Sklyanin Lax matrix

In this section, we construct an integrable time-discrete analogue of the anisotropic lattice Landau–Lifshitz model. To facilitate the computations, instead of canonical spins $\mathbf{S}$, it proves more convenient to employ an axially 'deformed spin' with components $\{\mathcal{K}, \mathcal{S}^+, \mathcal{S}^-\}$, enclosing a Poisson algebra

$$\mathcal{A}_q: \qquad \{\mathcal{K}, \mathcal{S}^\pm\} = \mp \mathrm{i}\varrho\, \mathcal{S}^\pm \mathcal{K}, \qquad \{\mathcal{S}^+, \mathcal{S}^-\} = -\frac{\mathrm{i}\varrho}{2}\left(\mathcal{K}^2 - \mathcal{K}^{-2}\right). \tag{8}$$

This Poisson algebra admits a quadric (Casimir) function

$$\mathcal{C}_0 \equiv \mathcal{S}^+ \mathcal{S}^- + \left(\frac{\mathcal{K} - \mathcal{K}^{-1}}{2}\right)^2, \tag{9}$$

satisfying $\{\mathcal{C}_0, \mathcal{S}^a\} = 0$. In terms of 'Sklyanin spins' [2], with components $\{\mathcal{S}^a\}_{a=0}^3$, with

$$\mathcal{S}^0 \equiv \frac{1}{2}(\mathcal{K} + \mathcal{K}^{-1}), \quad \mathcal{S}^1 \equiv \frac{1}{2}(\mathcal{S}^+ + \mathcal{S}^-), \quad \mathcal{S}^2 \equiv \frac{1}{2\mathrm{i}}(\mathcal{S}^+ - \mathcal{S}^-), \quad \mathcal{S}^3 \equiv \frac{1}{2}(\mathcal{K} - \mathcal{K}^{-1}), \tag{10}$$

the Hamiltonian of the lattice Landau–Lifshitz model takes a compact form

$$H_{\mathrm{LLL}}^\varrho \simeq \sum_{\ell=1}^L \log\left[\sum_{a=0}^3 g_a \mathcal{S}_\ell^a \mathcal{S}_{\ell+1}^a\right], \tag{11}$$

with couplings $g_0 = \sinh^2(\varrho)$, $g_1 = g_2 = 1$ and $g_3 = \cosh^2(\varrho)$.

We note that $\mathcal{A}_q$ is just the trigonometric limit of 'elliptic spins' that constitute the so-called quadratic *Poisson–Sklyanin algebra*, cf. Appendix A for further information. The algebra $\mathcal{A}_q$ becomes non-degenerate on symplectic leaves on which $\mathcal{C}_0$ takes a constant value $c_0$. We consider two *physically distinct* regimes:

(i) the *easy-axis* regime, with a real anisotropy (or interaction) parameter $\varrho \in \mathbb{R}_+$,

(ii) the *easy-plane* regime, with an interaction parameter $\gamma$ having a *compact support* [3] $\gamma \in [-\pi/2, \pi/2]$. The associated Hamiltonian is obtained from Eq. (11) by analytically continuing $\varrho$ from the real axis onto the imaginary axis, $\varrho \to \mathrm{i}\gamma$.

Our construction, which we outline in turn, applies to both of these regimes. Before that, we first briefly discuss the symplectic structure of the local space space. To obtain the easy-axis lattice Landau–Lifshitz model, the Casimir function $\mathcal{C}_0$ has to be fixed to

$$c_0 = \sinh^2(\varrho). \tag{12}$$

In this way, we select a two-dimensional non-degenerate Poisson submanifold $\mathcal{M}_{\mathrm{ea}}$ inside $\mathbb{R}^3$ which is diffeomorphic to a 2-sphere. In the easy-plane regime, obtained under analytic continuation $\varrho \to \mathrm{i}\gamma$, the submanifold $\mathcal{M}_{\mathrm{ep}}$ is also topologically equivalent to a 2-sphere provided $\gamma \in [-\pi/2, \pi/2]$. The upshot here is that there exist a smooth bijective mapping to the symplectic sphere $S^2$, implying that variables $\mathcal{K}$ and $\mathcal{S}^\pm$ can be parameterized in terms of classical *canonical* spins $\mathbf{S}$. A local mapping from $S^2$ to $\mathcal{M}_{\mathrm{ea}}$ is given by [60]

$$\mathcal{K} = e^{\varrho S^3}, \qquad \mathcal{S}^\pm = F_\varrho(S^3) S^\pm, \tag{13}$$

---

[2] Classical canonical spins can be recovered in the $\varrho \to 0$ limit by simultaneously rescaling $\mathcal{S}^a \to \mathcal{S}^a/\varrho$ for $a \in \{1, 2, 3\}$.

[3] This restriction has been overlooked in the previous works [70, 71] on the lattice Landau–Lifshitz model in continuous time, which specialize to the value $\gamma = 1$.

where

$$F_\varrho(s) \equiv \sqrt{\frac{\sinh^2(\varrho) - \sinh^2(\varrho s)}{1 - s^2}}, \tag{14}$$

the form of which follows from the Casimir function, see Eqs. (9),(12). Two imporatnt remarks are in order at this stage: (I) in the *easy-plane regime* (with $\varrho \to i\gamma$ and $F_{i\gamma}(s)$), Eq. (13) provides a *bijective* map only inside a compact interval of anisotropies $|\gamma| \leq \pi/2$; (II) there exist other Poisson submanifolds (classified in [60, 72]) where the classical Sklyanin algebra becomes non-degenerate. Those are not expressible in terms of canonical spins and thus we do not consider them here.

The discrete zero-curvature condition (3) may be formally viewed as a matrix refactorization problem. The task ahead of us is to find its explicit solution, i.e. expressing the output (primed) variables in terms of the input spin variables. We achieve this by first casting the Sklyanin Lax matrix in the form

$$L(z) = \frac{2}{z - z^{-1}} \begin{bmatrix} \frac{1}{2}\left(z\,\mathcal{K} - z^{-1}\mathcal{K}^{-1}\right) & \mathcal{S}^- \\ \mathcal{S}^+ & -\frac{1}{2}\left(z^{-1}\mathcal{K} - z\,\mathcal{K}^{-1}\right) \end{bmatrix}, \tag{15}$$

where we have for convenience introduced a multiplicative spectral parameter $z \in \mathbb{C}$,

$$z(\lambda) = e^{i\varrho\lambda}. \tag{16}$$

In Appendix A we detail out how the above Sklyanin Lax matrix arises from the limit of the quantum Lax operator associated to the quantum Yang–Baxter algebra.

The task of refactorization is unfortunately not straightforward. To begin with, Eq. (3) merely provides an implicit rule for the time-propagated input variables, and there is no obvious way of recasting it as an explicit map $\Phi$. [4] This difficulty can be elegantly overcome by exploiting certain factorization properties of the Lax matrix into more fundamental constituents.

## 2.2 Factorization

In this section, we outline how the Lax matrix can be further decomposed into more elementary constituents. Such factorization properties are indeed deeply rooted in the algebraic properties of certain types of Hopf algebras that are associated to simple Lie algebras or deformations thereof. For instance, it is well known that the universal *quantum* $\mathcal{R}$-matrix admits a triangular (Borel) decomposition [73–76]. The resulting 'partonic' Lax operators can be used to construct the Baxter $Q$-operators [77]. There exist various equivalent constructions in the literature, see e.g. Refs. [78–80] and Refs. [81–83]. We in turn demonstrate how the entire construction quite naturally elevates to the classical level by considering its semiclassical limit (outlined in Appendix A). Below we collect the relevant formulae without delving too much into formal considerations.

**Weyl variables.** To facilitate the factorization of the Lax matrix, we parametrize Sklyanin variables in terms of a classical Weyl–Poisson algebra. The latter is spanned by a pair of generators, $x, y$, satisfying the multiplicative canonical Poisson bracket

$$\{x, y\} = i\varrho\, x\, y, \tag{17}$$

---

[4]This is to some extent true even in the simplest isotropic case, where nonetheless it is possible to make use of vector identities to solve the zero-curvature condition explicitly.

via the prescription

$$\mathcal{K} = \nu y, \qquad \mathcal{S}^+ = \frac{x}{2}\left(\nu^2 y - \nu^{-2} y^{-1}\right), \qquad \mathcal{S}^- = -\frac{1}{2x}\left(y - y^{-1}\right), \tag{18}$$

where we have simultaneously defined, for compactness of notation, $\nu \equiv e^\varrho$. We accordingly make the identification

$$y = \nu^{-1} \mathcal{K} = e^{\varrho\,(S^3 - 1)}. \tag{19}$$

This implies that $y$ is real and positive in the easy-axis case, whereas in the easy-plane regime it becomes unimodular. In Weyl variables, however, the reality condition $\mathcal{S}^+ = \overline{\mathcal{S}^-}$ is not manifestly satisfied. To enforce it, the variable $x$ has to be restricted to a 'physical' submanifold by demanding the squared modulus to obey

$$|x|^2 = \begin{cases} -[y]_{\nu^2}, & y \in \mathbb{R}_+ \\ +[y]_{\nu^2}, & y \in U(1), \end{cases} \qquad [y]_{\nu^2} \equiv \frac{y - y^{-1}}{\nu^2 y - \nu^{-2} y^{-1}}. \tag{20}$$

**Factorization.** By introducing[5]

$$L_{x,y} \equiv \frac{1}{2}\left(z - z^{-1}\right) L(z), \tag{21}$$

along with a pair of spectral parameters,

$$u = \nu z, \qquad v = \nu z^{-1}, \tag{22}$$

satisfying $uv = \nu^2$, $u/v = z^2$, the Lax operator from Eq. (15) can be factorized as

$$L_{x,y}(V, U) \equiv \frac{1}{2} X^{-1} V Y U^T X = \frac{1}{2}\begin{bmatrix} uy - u^{-1}y^{-1} & -(y - y^{-1})x^{-1} \\ (uvy - u^{-1}v^{-1}y^{-1})x & -vy + v^{-1}y^{-1} \end{bmatrix}, \tag{23}$$

in terms of diagonal 'coordinate' matrices $X$ and $Y$

$$X = \begin{bmatrix} x & 0 \\ 0 & 1 \end{bmatrix}, \qquad Y = \begin{bmatrix} y & 0 \\ 0 & y^{-1} \end{bmatrix}, \tag{24}$$

and the 'spectral' matrices $U$ and $V$

$$U = \begin{bmatrix} u & -u^{-1} \\ -1 & 1 \end{bmatrix}, \qquad V = \begin{bmatrix} 1 & 1 \\ v & v^{-1} \end{bmatrix}. \tag{25}$$

**Elementary exchange relations.** We next describe various elementary exchange procedures involving $L_{x,y}(U, V)$. Firstly, interchanging the spectral matrices

$$L_{x,y}(V, U) = L_{x',y'}(U, V), \tag{26}$$

specifies a map $(x, y) \mapsto (x', y')$. The canonical Weyl relations are preserved [6] by additionally assuming $y' = y$, in which case

$$X^{-1} V Y U^T X = X'^{-1} U Y V^T X'. \tag{27}$$

---

[5]An overall multiplicative scalar factor bears no consequence on the dynamics as it cancels out from the discrete zero-curvature condition (3).

[6]This can be inferred directly using that $x' = x f(y)$ and $y' = y$ preserve the Poisson bracket (17).

Secondly, there exist quadratic exchange relations of the type

$$L_{x_1,y_1}(U_1,V_1)L_{x_2,y_2}(V_2,U_2) = L_{x_1',y_1'}(U_1,V_2)L_{x_2',y_2'}(V_1,U_2), \tag{28}$$

$$L_{x_1,y_1}(V_1,U_1)L_{x_2,y_2}(U_2,V_2) = L_{x_1',y_1'}(V_1,U_2)L_{x_2',y_2'}(U_1,V_2), \tag{29}$$

which are canonical provided that $x_1 = x_1'$, $x_2' = x_2$, implying

$$Y_1 V_1^T X_1 X_2^{-1} V_2 Y_2 = Y_1' V_2^T X_1 X_2^{-1} V_1 Y_2', \tag{30}$$

$$Y_1 U_1^T X_1 X_2^{-1} U_2 Y_2 = Y_1' U_2^T X_1 X_2^{-1} U_1 Y_2'. \tag{31}$$

From Eq. (27) we can deduce the following transformation of $(x,y) \mapsto (x', y'=y)$:

$$x' = -x\frac{uvy - (uvy)^{-1}}{y - y^{-1}}. \tag{32}$$

Equations (30) and (31) are equivalent under the interchange $u \leftrightarrow v$, so it suffices to solve only one of them. By direct calculation, exchanging $V_1$ and $V_2$ according to Eq. (30) yields

$$(y_1')^2 = y_1^2 \frac{v_1}{v_2}\frac{v_1 x_2 + v_2 x_1}{v_1 x_1 + v_2 x_2}, \qquad (y_2')^2 = y_2^2\frac{v_2}{v_1}\frac{v_1 x_1 + v_2 x_2}{v_1 x_2 + v_2 x_1}, \qquad x_\ell' = x_\ell, \tag{33}$$

$\ell \in \{1,2\}$, while analogously the exchange of $U_1$ and $U_2$ according to Eq. (31) yields the same expression with $v_\ell$ replaced by $u_\ell$:

$$(y_1')^2 = y_1^2\frac{u_1}{u_2}\frac{u_1 x_1 + u_2 x_2}{u_2 x_1 + u_1 x_2}, \qquad (y_2')^2 = y_2^2\frac{u_2}{u_1}\frac{u_2 x_1 + u_1 x_2}{u_1 x_1 + u_2 x_2}, \qquad x_\ell' = x_\ell. \tag{34}$$

By the same reasoning as previously (only swapping the roles of $x_\ell$ and $y_\ell$), transformations (33) and (34) are both canonical.

**Composite exchange relations.** By using the above elementary exchange relations it is possible to also exchange $U_1$ with $U_2$ between a pair of Lax matrices

$$L_{x_1,y_1}(V_1,U_1)L_{x_2,y_2}(V_2,U_2) = L_{x_1',y_1'}(V_1,U_2)L_{x_2',y_2'}(V_2,U_1). \tag{35}$$

This can be achieved by a composition of three elementary exchanges

$$(V_1,U_1),(V_2,U_2) \to (V_1,U_1),(U_2,V_2) \to (V_1,U_2),(U_1,V_2) \to (V_1,U_2),(V_2,U_1). \tag{36}$$

In an analogous fashion we can exchange $V_1$ with $V_2$

$$L_{x_1,y_1}(V_1,U_1)L_{x_2,y_2}(V_2,U_2) = L_{x_1',y_1'}(V_2,U_1)L_{x_2',y_2'}(V_1,U_2), \tag{37}$$

via the following sequence of elementary exchanges

$$(V_1,U_1),(V_2,U_2) \to (U_1,V_1),(V_2,U_2) \to (U_1,V_2),(V_1,U_2) \to (V_2,U_1),(V_1,U_2). \tag{38}$$

**Solving the zero-curvature condition.** To obtain the two-body propagator $\Phi$ the discrete zero-curvature condition, Eq. (3),

$$L_{x_1,y_1}(V_1,U_1)L_{x_2,y_2}(V_2,U_2) = L_{x_1',y_1'}(V_2,U_2)L_{x_2',y_2'}(V_1,U_1), \tag{39}$$

can now be solved by composing a '$U$-exchange' (35) with a '$V$-exchange' (37), in whichever order

$$(V_1,U_1),(V_2,U_2) \to (V_1,U_2),(V_2,U_1) \to (V_2,U_2),(V_1,U_1). \tag{40}$$

Their composition gives an explicit propagator $\Phi$ that solves the zero-curvature condition (39). Because the elementary exchanges are canonical transformations, the resulting propagator is automatically canonical. Explicit formulas for these compositions are rather cumbersome and we display them in Appendix B.

## 2.3 Two-body propagator

As in the isotropic case [55], the discrete 'Trotter' time-step $\tau \in \mathbb{R}$ enters the construction through the difference of the additive alternating (i.e. staggered) spectral parameters $\lambda^{\pm} \equiv \lambda \pm \tau/2$, carried by the light-cone Lax matrices, cf. Eq. (3). In terms of spectral parameter $u$ and $v$, this amounts to setting

$$u^{\pm} = v\, e^{\mathrm{i}\varrho\, \lambda^{\pm}}, \qquad v^{\pm} = v\, e^{-\mathrm{i}\varrho\, \lambda^{\pm}}, \tag{41}$$

recalling that $v = e^{\varrho}$. With the above prescription, the exchange relations (given by equations (35) and (37)) and the propagator $\Phi$ simplify considerably. Explicit expressions can be found in Appendix B).

The elementary propagator $\Phi_{\tau,\varrho}$ can alternatively be written explicitly using Sklyanin variables $\mathcal{K}, \mathcal{S}^{\pm}$. With aid of Eqs. (18) and some exercise, we find (where $w = e^{\mathrm{i}\varrho\, \tau}$)

$$\left(\frac{\mathcal{K}_1'}{\mathcal{K}_1}\right)^2 = \frac{(v^2 + v^{-2})\big(\mathcal{K}_1 \mathcal{K}_2 + \mathcal{K}_1^{-1}\mathcal{K}_2^{-1}\big) - 2\mathcal{K}_1 \mathcal{K}_2^{-1} - (w^2 + w^{-2})\mathcal{K}_1^{-1}\mathcal{K}_2 + 4w\mathcal{S}_1^+ \mathcal{S}_2^- + 4w^{-1}\mathcal{S}_1^- \mathcal{S}_2^+}{(v^2 + v^{-2})\big(\mathcal{K}_1 \mathcal{K}_2 + \mathcal{K}_1^{-1}\mathcal{K}_2^{-1}\big) - 2\mathcal{K}_1^{-1}\mathcal{K}_2 - (w^2 + w^{-2})\mathcal{K}_1 \mathcal{K}_2^{-1} + 4w\mathcal{S}_1^- \mathcal{S}_2^+ + 4w^{-1}\mathcal{S}_1^+ \mathcal{S}_2^-},$$

$$\left(\frac{\mathcal{K}_2'}{\mathcal{K}_2}\right)^2 = \frac{(v^2 + v^{-2})\big(\mathcal{K}_1 \mathcal{K}_2 + \mathcal{K}_1^{-1}\mathcal{K}_2^{-1}\big) - 2\mathcal{K}_1^{-1}\mathcal{K}_2 - (w^2 + w^{-2})\mathcal{K}_1 \mathcal{K}_2^{-1} + 4w\mathcal{S}_1^- \mathcal{S}_2^+ + 4w^{-1}\mathcal{S}_1^+ \mathcal{S}_2^-}{(v^2 + v^{-2})\big(\mathcal{K}_1 \mathcal{K}_2 + \mathcal{K}_1^{-1}\mathcal{K}_2^{-1}\big) - 2\mathcal{K}_1 \mathcal{K}_2^{-1} - (w^2 + w^{-2})\mathcal{K}_1^{-1}\mathcal{K}_2 + 4w\mathcal{S}_1^+ \mathcal{S}_2^- + 4w^{-1}\mathcal{S}_1^- \mathcal{S}_2^+},$$

$$\begin{bmatrix} (\mathcal{S}_1^{\pm})' \\ (\mathcal{S}_2^{\pm})' \end{bmatrix} = \Omega(w^{\pm 1}) \begin{bmatrix} \mathcal{S}_1^{\pm} \\ \mathcal{S}_2^{\pm} \end{bmatrix}, \tag{42}$$

with

$$\Omega(w) = \frac{1}{\mathcal{K}_1 \mathcal{K}_2 w - (\mathcal{K}_1 \mathcal{K}_2 w)^{-1}} \begin{bmatrix} \mathcal{K}_1' \mathcal{K}_2 - (\mathcal{K}_1' \mathcal{K}_2)^{-1} & (\mathcal{K}_1'/\mathcal{K}_1)w - (\mathcal{K}_1/\mathcal{K}_1')w^{-1} \\ (\mathcal{K}_2'/\mathcal{K}_2)w - (\mathcal{K}_2/\mathcal{K}_2')w^{-1} & \mathcal{K}_1' \mathcal{K}_2 - (\mathcal{K}_1' \mathcal{K}_2)^{-1} \end{bmatrix}. \tag{43}$$

We remind the reader that in the easy-axis regime $\mathcal{K} \in \mathbb{R}_+$, whereas in the easy-plane regime $\gamma S^3 = \arg(\mathcal{K}) \in [-\pi/2, \pi/2]$. Therefore, there is no ambiguity in Eqs. (42) when taking the square roots to determine $\mathcal{K}_\ell'$.

**Properties.** There are a number of important properties worth highlighting:

- In the continuous-time limt $\tau \to 0$, the propagator $\Phi_{\tau,\varrho}$ reduces to the identity map.

- The mapping (42) manifestly preserves the product $\mathcal{K}_1 \mathcal{K}_2$, implying that the local propagator conserves the third component of the spin,

$$S_1^3 + S_2^3 = (S_1^3)' + (S_2^3)'. \tag{44}$$

  This local $U(1)$ symmetry directly implies conservation of $S_{\mathrm{tot}}^3 \equiv \sum_{\ell=1}^{L} S_\ell^3$ under the full propagator $\Phi^{\mathrm{full}}$ which, in the Hamiltonian limit, $\tau \to 0$, becomes the Noether charge of the uniaxially anisotropic Landau–Lifshitz model.

- In the easy-axis regime, the local propagator (42) is a *periodic* function of interaction $\varrho$, with period $2\pi/\tau$.

- The two-body propagator $\Phi_{\tau,\varrho}$ is invariant under the transformation

$$\varrho \to -\varrho, \quad \tau \to -\tau, \qquad \mathbf{S}_{1,2} \to -\mathbf{S}_{1,2}. \tag{45}$$

  It is therefore sufficient to consider only the interval $\varrho \in [0, \pi/\tau]$.

- In the *easy-axis regime*, at $\varrho = \pi/\tau$ (equiv. $w = -1$) the dynamics trivializes to the identity for all $\tau$,

$$\lim_{\varrho \to \pi/\tau} \Phi_{\tau,\varrho}(\mathbf{S}_1, \mathbf{S}_2) = (\mathbf{S}_1, \mathbf{S}_2). \tag{46}$$

- In the *easy-plane regime*, the dynamics is *not* periodic in $\varrho$, but is instead confined to the interval $\gamma \in [-\pi/2, \pi/2]$. By virtue of Eqs. (45), it is sufficient to consider only the range $\gamma \in [0, \pi/2]$.

- In the easy-plane regime, in the limit $\tau \to \infty$ with $\gamma$ fixed (i.e. $|w| \to \infty$), $\Phi_{\tau,\mathrm{i}\gamma}$ reduces to a permutation

$$\lim_{\tau \to \infty} \Phi_{\tau,\mathrm{i}\gamma}(\mathbf{S}_1, \mathbf{S}_2) = (\mathbf{S}_2, \mathbf{S}_1). \tag{47}$$

- The two-body propagator $\Phi$ is a *classical Yang–Baxter map* [66, 84–86], obeying the braided Yang–Baxter relation on the Cartesian product $\mathcal{M}_1^{\times 3}$ (suppressing the anisotropy parameter)

$$\Phi_{\lambda_2-\lambda_3}^{(1,2)} \circ \Phi_{\lambda_1-\lambda_3}^{(2,3)} \circ \Phi_{\lambda_1-\lambda_2}^{(1,2)} = \Phi_{\lambda_1-\lambda_2}^{(2,3)} \circ \Phi_{\lambda_1-\lambda_3}^{(1,2)} \circ \Phi_{\lambda_2-\lambda_3}^{(2,3)}, \tag{48}$$

where $\Phi^{(i,j)}$ signifies that $\Phi$ operates non-identically on $i$-th and $j$-th copy of $\mathcal{M}_1^{\times 3}$.

**Discrete Noether current.**    The lattice Landau–Lifshitz model, cf. Eq. (11), is invariant under global rotations about the third axis. This imples, by virtue of the Noether theorem, a globally conserved Noether charge (magnetization) in the system. We have just explained, cf. Eq. (44), that the same property holds even in the discrete-time setting.

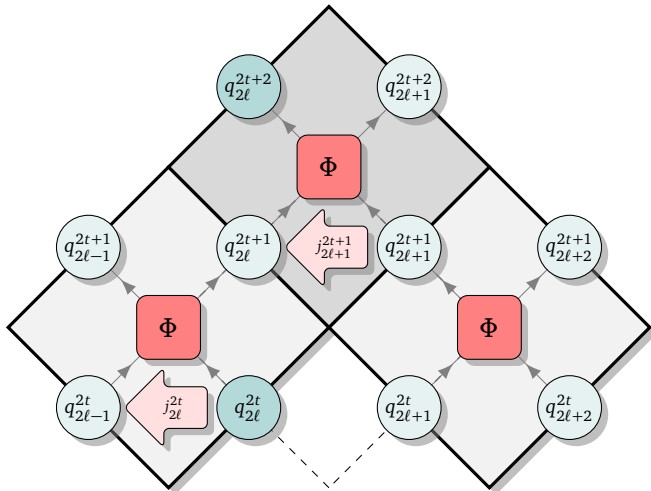

Figure 3: Discrete Noether current, shown at an even site $2\ell$: the time-increment $t \to t+2$ of charge density, $(q_{2\ell}^{t+2} - q_{2\ell}^{t})/\tau$ equals the difference of two nearby current densities at consecutive times, $j_{2\ell+1}^{2t+1} - j_{2\ell}^{2t}$. Current densities $j_\ell^t$, depicted by pink arrows, can be compactly expressed as the difference of charge densities at consecutive times, $j_\ell^t = (q_\ell^{t+1} - q_\ell^t)/\tau$.

By virtue of the global $U(1)$ symmetry, the dynamics admits a conserved Noether current. Its spatial component, i.e. the 'charge density', is given by the projection of $\mathbf{S}$ onto the third axis, $q = S^3$. The associated temporal component is the 'current density' which we subsequently denote by $j$. Owing to the even-odd staggered structure of the full propagator $\Phi^{\mathrm{full}}$, the discrete continuity equations at even and odd sites are of the form

$$\frac{1}{\tau}\left(q_{2\ell}^{2t+2} - q_{2\ell}^{2t}\right) + j_{2\ell+1}^{2t+1} - j_{2\ell}^{2t} = 0, \qquad \frac{1}{\tau}\left(q_{2\ell+1}^{2t+2} - q_{2\ell+1}^{2t}\right) + j_{2\ell+2}^{2t} - j_{2\ell+1}^{2t+1} = 0, \tag{49}$$

respectively. While the charge densities $q_\ell^t \equiv q(\ell, t)$ are ultralocal, namely they sit at site $\ell$ and

time $t$, the associated current density corresponds to their forward differences [7]

$$j_\ell^t \equiv j(\ell, t) = \frac{1}{\tau}\big(q(\ell, t+1) - q(\ell, t)\big),\tag{50}$$

which can be immediately verified by plugging the expression into the discrete continuity equations (49) and using the conservation property (44). Notice that $q_\ell^{t+1}$ is a function of two adjacent charge densities at the previous time slice, namely $q_\ell^t$ and $q_{\ell\pm1}^t$, cf. Figure 3, that enter the local propagator $\Phi$.

**Isotropic, continuous time and field-theory limits.** The two-parametric dynamical map $\Phi_{\tau,\varrho}^{\text{full}}$ admits various important limits: (i) the limit of vanishing anisotropy $\varrho \to 0$, (ii) the continuous-time limit $\tau \to 0$ and (iii) the continuous space-time (i.e. field-theory) limit by additionally taking the long wavelength limit. The resulting models, outlined in Appendix D, can be all regarded as members of the 'Landau-Lifshitz family'.

# 3 Magnetization transport

We demonstrate the utility of the constructed dynamical system by studying magnetization transport in thermal equilibrium. We focus on a numerical computation of the linear transport coefficients that quantify magnetization transport on a large spatio-temporal (i.e. hydrodynamic) scale, namely the spin Drude weight and the spin diffusion constant. After introducing the equilibrium measure and transport coefficients we proceed by systematically analyzing the easy-axis and the easy-plane regimes.

## 3.1 Transport coefficients

**Equilibrium ensemble.** We are interested in the dynamical response in our model at finite magnetization density. To this end, we introduce a one-parametric family of local measures $\rho_1(\mu)$ on $\mathcal{M}_1$,

$$\rho_1(\mathbf{S}) = \frac{1}{4\pi}\frac{\mu}{\sinh\mu}e^{\mu S^3},\tag{51}$$

which can be interpreted as the grand-canonical measure with chemical potential $\mu$. [8] We accordingly define a separable *stationary* measure $\rho_L \equiv \rho_L\big(\{\mathbf{S}_\ell\}_{\ell=1}^L\big)$ on the full phase space $\mathcal{M}_L$,

$$\rho_L = \prod_{\ell=1}^L \rho_1(\mathbf{S}_\ell),\tag{52}$$

with chemical potential $\mu$ enforcing a non-zero average of magnetization. Invariance of $\rho_L$ under the full dynamics $\Phi^{\text{full}}$ follows from the fact that the quadratic Sklyanin bracket is preserved by the action of $\Phi$ (see Ref. [53] for an analogous formal proof).

We now pass over to the thermodynamic limit by sending $L \to \infty$. The grand-canonical free energy per site is given by

$$f(\mu) = -\log \mathcal{Z}_1(\mu),\tag{53}$$

---

[7]We note that the current density is only defined up to a total difference. We fix this gauge freedom by demanding that the current vanishes when both input spins are equal.

[8]Such a one-body measure maximizes the Shannon (or Gibbs) entropy of a state, that is $-\int d\Omega \rho_1(\mathbf{S})\log\rho_1(\mathbf{S})$ where $d\Omega$ is a uniform measure on $S^2$ with a prescribed average $\langle S^3 \rangle$ set by a Lagrange multiplier $\mu$. In the continuous-time (i.e. Hamiltonian) limit, $\tau \to 0$, $\rho_1(\mathbf{S})$ is the grand-canonical Gibbs measure in the limit of infinite temperature.

where

$$\mathcal{Z}_1(\mu) = \int_{S^2} d\Omega\, \rho_1(\mathbf{S}) = \frac{4\pi \sinh\mu}{\mu}, \tag{54}$$

represents the onsite grand-canonical partition function, with $d\Omega$ denoting the volume element on $S^2$. Writting $d\Omega^{\text{full}} \equiv \prod_{\ell=1}^{L} d\Omega(\mathbf{S}_\ell)$, the corresponding thermodynamic average of any local observable $\mathcal{O}$ is therefore computed as

$$\langle \mathcal{O} \rangle_\mu = \lim_{L\to\infty} \left[ \mathcal{Z}_1^L(\mu) \right]^{-1} \int_{\mathcal{M}_L} d\Omega^{\text{full}} \rho_L\, \mathcal{O}. \tag{55}$$

For example, the average of magnetization density reads

$$\langle S^3 \rangle_\mu = \coth\mu - 1/\mu. \tag{56}$$

In the following we accordingly take $L \to \infty$ and operate in the thermodynamic setting.

**Conductivity.**  In the linear-reponse regime, transport coefficients are encoded in the real part of the frequency-dependent conductivity $\sigma(\omega)$, which is in general decomposed as

$$\sigma(\omega) = \pi\mathcal{D}\,\delta(\omega) + \sigma_{\text{reg}}(\omega). \tag{57}$$

The $\delta$-peak owes its existence to the presence of long-lived excitations which ballistically transport the charges through the system, characterized by a coefficient $\mathcal{D}$ called the Drude weight. The d.c. component of the 'regular' part, that is $\sigma_{\text{reg}}(0)$, characterizes transport on sub-ballistic scales. Provided $\sigma_{\text{reg}}(0)$ is finite, it can be associated to the spin diffusion constant defined as

$$D = \chi^{-1} \lim_{\omega\to 0} \sigma_{\text{reg}}(\omega), \tag{58}$$

where $\chi$ denotes static spin susceptibility, $\chi(\mu) = \partial^2 f(\mu)/\partial^2\mu = 1 + \mu^{-2} - \coth^2\mu$. In integrable *interacting* systems, the diffusion constant quantifies the diffusive spreading which quasiparticle excitations exhibit upon ellastic collisions (see e.g. Refs. [40,87]). As customary, we define the dynamical structure factor

$$S(\ell, t) = \langle q(\ell, t) q(0, 0) \rangle_\mu^c, \tag{59}$$

with a sum rule $\sum_{\ell \in \mathbb{Z}} S(\ell, t) = \chi$. The spin Drude weight governs the time-asymptotic growth of its second moment

$$\sum_{\ell \in \mathbb{Z}} \ell^2 S(\ell, t) \simeq \mathcal{D}\, t^2 + \mathcal{O}(t), \tag{60}$$

where the subleading correction, which equals $2\chi D$, stores information about the spin diffusion constant $D$. We note that numerically extracting $D$ from the $\mathcal{O}(t)$ broadening of the second moment (60) has not proven particularly reliable.

We shall employ an alternative approach and define the (linear) transport coefficients through the Kubo formula. The extensive magnetization current $J$ at physical times $2t$ follows directly from the discrete continuity equation (49) and is given by

$$J(2t) = \sum_{\ell \in \mathbb{Z}} \left( j(2\ell, 2t) + j(2\ell + 1, 2t + 1) \right). \tag{61}$$

The spin Drude weight corresponds to the asymptotic value of the current autocorrelation function

$$\mathcal{D} = \lim_{t\to\infty} \frac{1}{t} \sum_{t'=0}^{t} \langle J(2t') j(0, 0) \rangle_\mu^c, \tag{62}$$

whereas the spin diffusion constant is the integrated current autocorrelator with the asymptotic value subtracted

$$D = \chi^{-1} \lim_{t \to \infty} \tau \sum_{t'=0}^{t} \left( \langle J(2t')j(0,0) \rangle_\mu^c - \mathcal{D} \right). \tag{63}$$

Lastly, we introduce the dynamical exponent $\alpha$ that characterizes the late-time decay of the dynamical spin structure factor

$$S(0,t) \sim t^{-\alpha}. \tag{64}$$

Ballistic spreading corresponds to exponent $\alpha = 1$. On the other hand, we can speak of normal diffusion when (i) $\alpha = 1/2$ and (ii) the dynamical structure factor converges asymptotically to a Gaussian stationary scaling profile,

$$S(\ell, t) \simeq \frac{\chi}{\sqrt{2\pi Dt}} e^{-\ell^2/2Dt}. \tag{65}$$

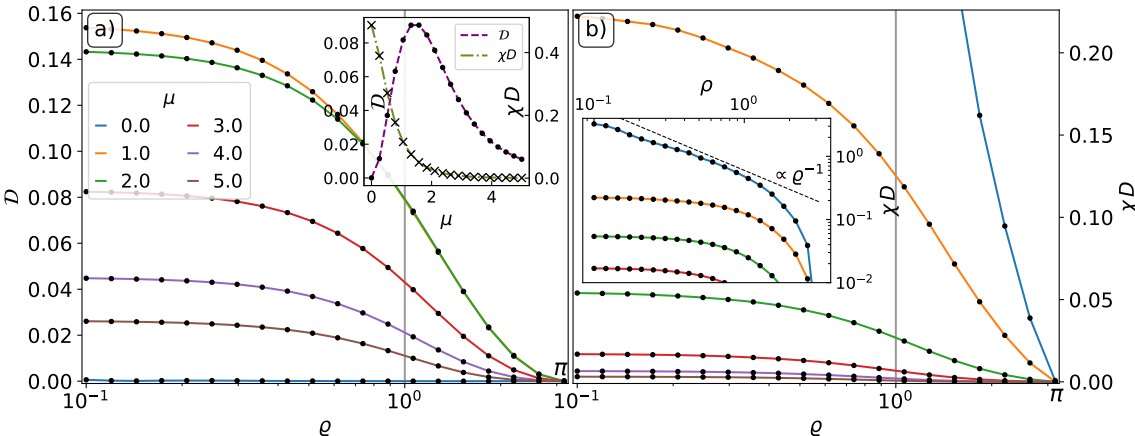

Figure 4: (a) Spin Drude weight $\mathcal{D}$ and (b) diffusion constant $D$ in the discrete Landau–Lifshitz model, cf. Eq. 4, in the easy-axis regime, depending on the chemical potential $\mu$ and the anisotropy $\varrho$. (a, inset) Drude weight (purple) and diffusion constant (olive green) as functions of $\mu$ at $\varrho = 1$ (vertical grey line). (b, inset) Divergence of $D$ at zero average magnetization, $\mu = 0$ (blue curve), in the isotropic limit $\varrho \to 0$, compared to $\sim 1/\varrho$ power-law scaling (black dashed line). Simulation parameters: $\tau = 1$, $L = 2^{12}$, averaged over $10^5$ initial configurations.

## 3.2 Numerical simulations: easy-axis regime

We first focus our analysis on the spin Drude weight and diffusion in the easy-axis regime. The results are shown in Figure 4. The asymptotic stationary profiles and the numerically estimated dynamical exponents are shown in Figure 5.

**Drude weight.** The spin Drude weight is finite for all anisotropies $\varrho$ and arbitrary finite magnetization density $\mu > 0$. In the unmagnetized ensemble ($\mu = 0$), corresponding to a uniform measure $\rho_1 = 1/\text{Vol}(S^2) = 1/4\pi$, the Drude weight vanishes. This point is distinguished by the discrete $\mathbb{Z}_2$ symmetry, corresponding to the global reversal of all spins, $S_\ell^\pm \mapsto S_\ell^\mp, S_\ell^z \mapsto -S_\ell^z$, which is a canonical transformation, i.e. preserving the Poisson bracket. We find, moreover, that both transport coefficients vanish when $\varrho \to \pi/\tau$, consistent with the dynamics becoming trivial in that limit, cf. Eq. (46).

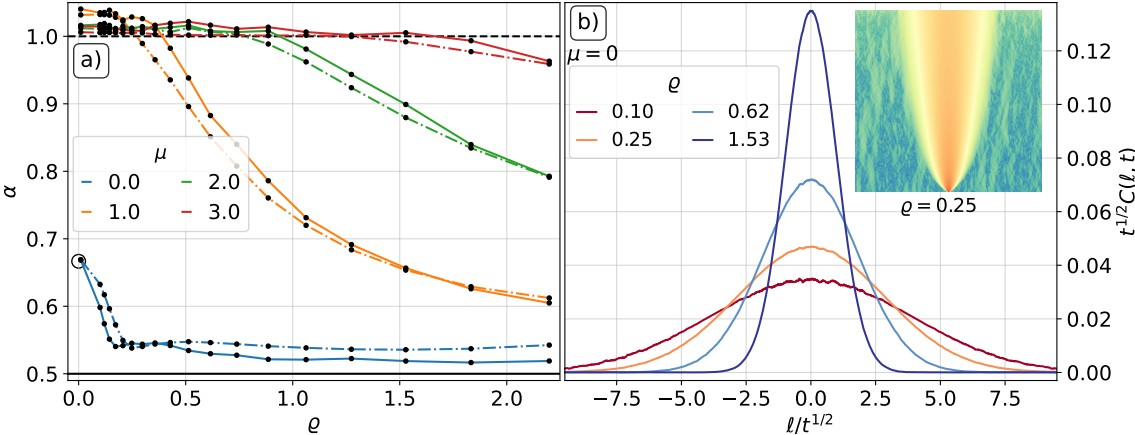

Figure 5: (a) Finite-time dynamical exponents $\alpha$ ($t \in [100, 300]$ dashed-dotted lines, $t \in [300, 1000]$ solid lines) depending on the anisotropy $\varrho$ at four values of the chemical potentials $\mu$. Magnetized ensembles ($\mu > 0$) exhibit ballistic transport with exponent $\alpha = 1$. The sub-leading diffusive correction increases with $\varrho$ and overwhelms the ballistic contribution at finite times. In the non-magnetized sector at $\mu = 0$, the dynamical exponent is approximately $\alpha \approx 2/3$ (black circle) when approaching $\varrho \rightarrow 0$, consistent with KPZ-type superdiffusion at the isotropic point $\varrho = 0$. At large values of $\varrho$, the dynamical exponent approaches $\alpha \approx 1/2$ which is indicative of normal diffusion. (b) Rescaled diffusive stationary scaling profiles of the dynamical structure factors $t^{1/2} C(\ell, t)$ at $\mu = 0$ and different values of $\varrho$. Simulation parameters: $\tau = 1$, $L = 2^{12}$, averaged over $10^5$ initial configurations.

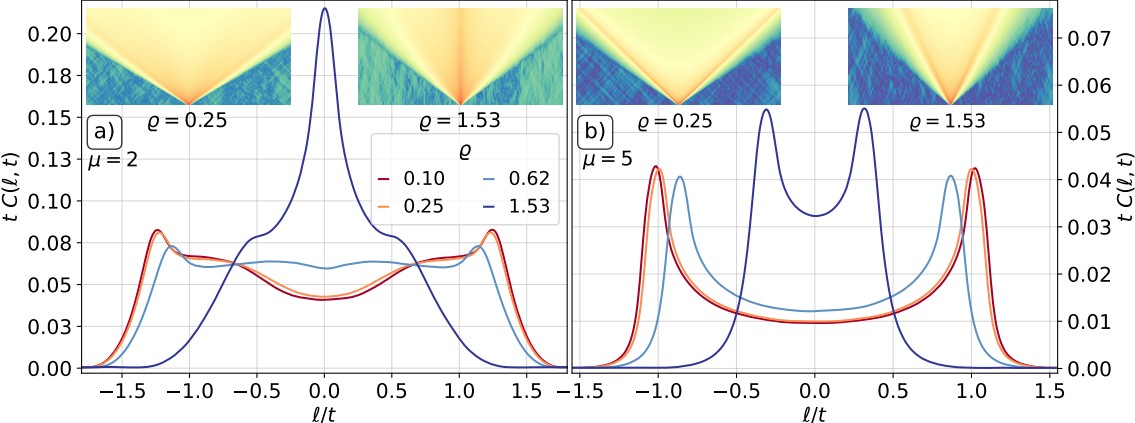

Figure 6: Ballistic stationary scaling profiles of dynamical structure factors $S(\ell, t)$ (shown in insets) computed at $\mu = 2, 5$. Simulation parameters: $\tau = 1, L = 2^{12}$, averaged over $10^5$ initial configurations.

Fixing anisotropy to $\varrho = 1$, we find a non-monotonic dependence of $\mathcal{D}$ on the chemical potential $\mu$, see the inset plot in Figure 4, which is consistent with the picture of dressed quasiparticles [40, 41]: in an unpolarized ensemble, the dressed magnetization vanishes and so does the spin Drude weight. For small $\mu$, the quasiparticles behave paramagnetically, with dressed magnetization behaving as $\sim \mu$. For strong polarizations, $\mu \rightarrow \infty$, one approaches a $\mathbb{Z}_2$-degenerate quasiparticle pseudovacuum where the quasiparticles become effectively free. In this regime, however, the quasiparticle density vanishes and the spin Drude weight drops to zero.

**Diffusion constant.** The spin diffusion constant $D$ is finite for all $\mu > 0$. We find that it decreases monotonically with increasing $\mu$. At any given anisotropy $\varrho$, $D$ grows upon lowering $\mu$, and attains its maximal value at $\mu = 0$. There, upon approaching the isotropic point, $\varrho \to 0$, the diffusion constant $D$ diverges in (approximately) algebraic fashion, $D \sim 1/\varrho^{\varkappa_{\mathrm{ea}}}$, as shown in Figure 4 (right inset). This conforms with a theoretical expectation $\varkappa_{\mathrm{ea}} = 1$ based on the known behavior in the quantum Heisenberg chain [34]. [9] A reliable estimation of $\varkappa_{\mathrm{ea}}$ from the numerical data, as shown in Figure 4 (b, inset, dashed black line), is however rather difficult due to a divergent relaxation timescale in the proximity of the isotropic point. Singularity of the spin diffusion constant signals the onset of *superdiffusion* [88] which has recently been under intense scrutiny [30, 32, 53, 55, 71, 89–91]. There exists ample numerical evidence [32, 33, 53, 55, 71, 92, 93] that this phenomenon belongs to the universality class of the Kardar–Parisi–Zhang (KPZ) equation [94]. In Figure 5, we numerically corroborate the predicted anomalous dynamical exponent exponent $\alpha = 2/3$.

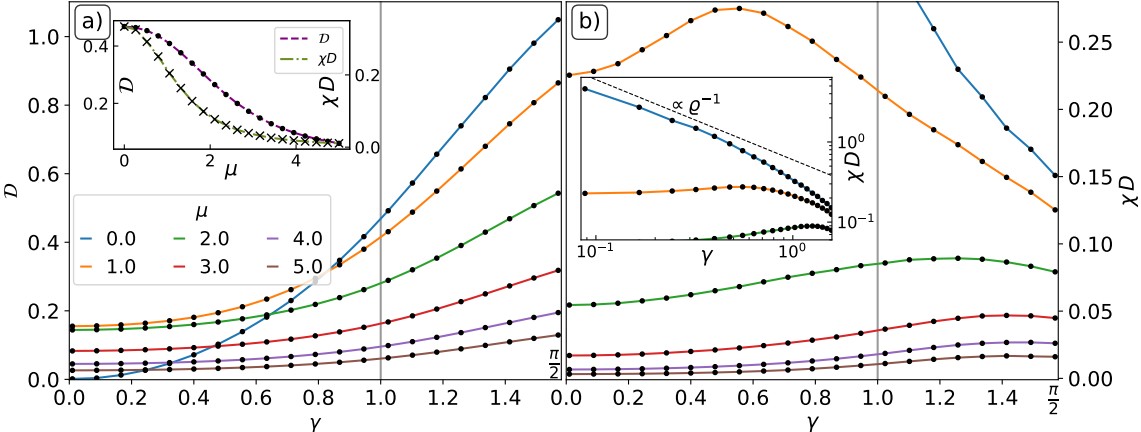

Figure 7: (a) Spin Drude weight $\mathcal{D}$ and (b) and diffusion constant $D$ in the discrete Landau–Lifshitz model, cf. Eq. (4), in the easy plane-regime, depending on chemical potential $\mu$ and anisotropy $\gamma$. (a, inset) Drude weight (purple) and diffusion constant (olive green) as functions of $\mu$ at $\gamma = 1$ (vertical grey line). (b, inset) Divergence of $D$ at zero average magnetization, $\mu = 0$ (blue curve), in the isotropic limit $\gamma \to 0$, compared to $\sim 1/\varrho$ power-law scaling (black dashed line). Simulation parameters: $\tau = 1$, $L = 2^{12}$, averaged over $10^5$ initial configurations.

The transient effects can be discerned by plotting the dynamical structure factors (the equal-time profiles of dynamical correlation functions of the spin density), as shown in Figure 5. While the dynamics is ballistic for all values of $\varrho > 0$ and $\mu > 0$, the finite-time dynamical exponent estimated based on (64) is found to be ballistic, $\alpha = 1$, only below a threshold value of $\varrho$ (which increases with increasing $\mu$). This is due to the fact that the value of $\mathcal{D}$ is small in comparison with $D$. By contrast, at $\mu = 0$, the estimated exponent is approximately diffusive, $\alpha \approx 1/2$ for $\varrho \gtrsim 1$, while when approaching $\varrho \to 0$ it starts to drift towards the expected superdiffusive exponent $\alpha = 2/3$. At $\mu = 0$, the stationary cross-sections shown in Figure 5 (b) are single-peaked, consistent with the absence of ballistic transport. As $\varrho \to 0$, the broadening of the central peak is compatible with the aforementioned divergence of the diffusion constant.

---

[9]In the gapped phase of the Heisenberg spin-1/2 chain with anisotropy $\Delta = \cosh(\eta)$, the spin diffusion constant diverges as $\sim 1/\eta$ as $\eta \to 0$. In the semiclassical limit, easy-axis anisotropy $\varrho$ is proportional to $\eta$, see Appendix A.

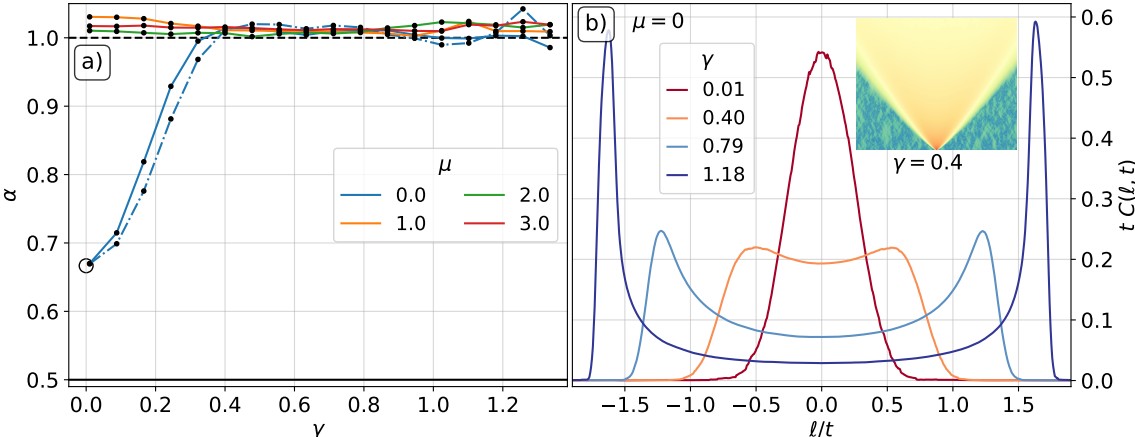

Figure 8: (a) Finite-time dynamical exponents $\alpha$ (full blue line for $t \in [300, 1000]$, blue dashed-dotted line for $t \in [100, 300]$) as functions of anisotropy for different values of chemical potentials $\mu$. At all $\mu > 0$, we find ballistic transport ($\alpha = 1$). At $\mu = 0$ and $\gamma \to 0$, the dynamical exponent crosses over to superdiffusion with $\alpha = 2/3$ (black circle). (b) Rescaled ballistic stationary profiles of the dynamical structure factor $t\, C(\ell, t)$ at $\mu = 0$.

## 3.3 Numerical simulations: easy-plane regime

We now repeat the above numerical analysis for the easy-plane regime. The spin Drude weights and diffusion constants are displayed in Figure 7, while the asymptotic stationary profiles of the dynamical structure factors, alongside the finite-time dynamical exponents, are shown in Figure 8 and Figure 9.

**Drude weight.** In the easy-plane regime, the spin Drude weight $\mathcal{D}$ takes a *non-zero* value for all values of $\mu$ and $\gamma$, except at the isotropic point at zero magnetization ($\mu = \gamma = 0$). Upon lowering $\gamma$, the spin Drude weight $\mathcal{D}$ now monotonously *decreases*, for any fixed $\mu$, to a limiting value at $\gamma = 0$ which is non-zero for $\mu \neq 0$.

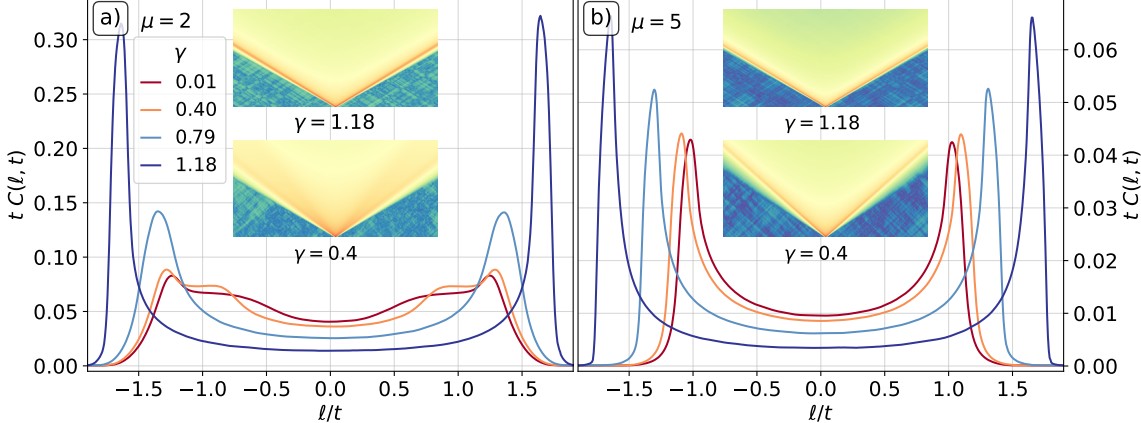

Figure 9: Rescaled ballistic stationary profiles of the dynamical structure factor $t\, C(\ell, t)$ for (a) $\mu = 2$ and (b) $\mu = 5$, with the corresponding space-time dependence $C(\ell, t)$ (inset plots). Simulation parameters: $\tau = 1$, $L = 2^{12}$, averaged over $10^5$ initial configurations.

It is worth highlighting that $\mathcal{D}$ does not vanish even in a non-magnetized ensemble, in

stark contrast with the easy-axis regime. Although this is perfectly aligned with previous numerical studies of the lattice Landau–Lifshitz model [70, 71], a proper theoretical explanation is currently still lacking. At this juncture, we can offer two complementary interpretations:

(a) from the quasiparticle point of view, finite spin Drude weight in a non-magnetized sector indicates that certain excitations attain finite dressed magnetization even in the $\mu \to 0$ limit, mirroring the situation in the easy-plane phase of the quantum Heisenberg XXZ spin chain [41, 95],

(b) from the viewpoint of the Mazur–Suzuki hydrodynamic projection [96, 97] (for more information see e.g. [87, 98, 99]) a finite value of $\mathcal{D}$ necessitates the existence of local (or quasilocal, see Refs. [98, 100, 101]) conserved quantities with finite spin-charge susceptibility even at $\mu \to 0$, that is $\langle J Q \rangle_0^c > 0$.

Conserved quantities with a finite overlap with spin current have indeed been found in the gapless phase of the quantum Heisenberg chain (see Refs. [102–105] and [101] for a review). They are commonly referred to as the 'Z-charges'. Presently however, the standard local conservation laws $Q_{k;\tau}^{\pm}$ extracted by expanding the trace of the monodromy matrix as an analytic series (see the derivation in Appendix C) all satisfy $\langle J Q_{k;\tau}^{\pm} \rangle_0^c = 0$. This indeed follows directly from a simple symmetry argument: while all local charges $Q_{k;\tau}$ remain invariant under the spin-reversal transformation, the spin current $J$ flips a sign.

**Diffusion constant.** The diffusion constant $D$ is finite for all values of $\gamma$ provided $\mu > 0$. In the $\mu \to 0$ limit however, it once again diverges as $D \sim 1/\gamma^{\varkappa_{\text{ep}}}$ algebraically with an estimated exponent $\varkappa_{\text{ep}} = 1$, cf. Figure 7 (b, inset). We can observe, see Figure 8 (a, blue line), a crossover behavior in the vicinity of the isotropic point from ballistic to superdiffusive dynamics. In the easy-plane regime, the stationary profiles (Fig. 9) have two peaks which are indicative of ballistic spreading. At large $\tau$ (with fixed $\gamma$) the two-particle map reduces to the permutation, see Eq. (47), meaning that the dynamics becomes non-interacting and hence the structure factor concentrates near the light-cone boundary peaked at $\ell/t = \pm 2$.

# 4 Conclusion

We have introduced a new integrable model in discrete space-time, representing classical spins which interact locally via an anisotropic spin-spin interaction. The model can be perceived as an integrable discrete-time analogue of the uniaxailly anisotropic Landau-Lifshitz field theory. The key ingredient of our algebraic construction is the discrete version of the zero-curvature condition on an auxiliary light-cone lattice, which implicitly defines the elementary time-propagator. We managed to express the elementary propagator explicitly as a two-body symplectic transformation by exploiting certain factorization properties of the Lax matrix. The full propagator takes the form of a classical circuit made of symplectic maps. The two free parameters of the model are the Trotter time-step and the interaction anisotropy, the latter can be chosen real or purely imaginary (corresponding to the easy-axis and easy-plane regimes, respectively).

Our discrete model provides an efficient explicit integration scheme that manifestly preserves integrability. As an important immediate physics application, we numerically studied magnetization transport in grand-canonical stationary ensembles at different values of magnetization density. With only a moderate computational effort, we compute the linear transport coefficients (the spin Drude weight and spin diffusion constant) with aid of the Kubo formula, both in the easy-axis and easy-plane regimes.

The spin Drude weight is a non-negative quantity. As expected, we found that it only vanishes in (a) the limit of strong polarization and in (b) the easy-axis regime at zero net magnetization. On sub-ballistic scales, there is a finite diffusive correction characterized by the spin diffusion constant. According to expectations, we have found that the spin diffusion constant blows up when approaching the isotropic point (within either regime). This divergence is numerically estimated to be roughly inversely proportionally to the anisotropy strength. Close to the isotropic point, we encountered discernible finite-time crossover effects which signal the onset of spin superdiffusion [41] (characterized by the dynamical exponent $\alpha = 2/3$).

In the easy-plane regime, the spin Drude weight retains a finite value even at zero net magnetization, consistent with the findings of Refs. [70, 71]. It is well-known indeed that the same behavior occurs in the the gapless phase of the Heisenberg quantum spin chain, where the non-vanishing of the spin Drude weight is intimately linked with the existence of the so-called quasilocal conservation laws [101–103] of odd parity under the spin-reversal transformation. The most suggestive explanation of this peculiarity is that there exist additional 'hidden' (quasi)local conservation laws in the classical spin model as well – a classical analogue of the $Z$-charges.

The hope is that our numerical data can serve as an accurate test of these theoretical predictions. The spin Drude weight and diffusion constant are in principle amenable to analytic computation, e.g. using the universal formulae for the Drude weights [106, 107] and diffusion constants [108–110] obtained within the formalism of generalized hydrodynamics [27, 28]. This computation however requires the knowledge of the quasiparticle content of the model and the associated thermodynamic state functions [87], which are yet to be derived, e.g. by means of the inverse scattering techiques [60, 64, 65, 111] adapted to the fully discrete setting. We leave this task for future research.

## Acknowledgements

The work has been supported by ERC Advanced grant 694544 – OMNES and the program P1-0402 of Slovenian Research Agency.

# Appendices

## A  Semiclassical limit of quantum algebra

The algebraic structures pertaining to the lattice Landau–Lifshitz model can be derived systematically by taking the semiclassical limit of a quasi-triangular Hopf algebra (i.e. the 'quantum group') $\mathcal{U}_q(\mathfrak{su}_2)$. For the reader's convenience, we summarize the key steps of the derivation.

**Fundamental commutation relation.**   A natural starting point is to consideer the so-called fundamental commutation (or RLL) relation of the difference form [69, 73, 112]

$$R_{12}(\lambda - \lambda')\hat{L}_{13}(\lambda)\hat{L}_{23}(\lambda') = \hat{L}_{23}(\lambda')\hat{L}_{13}(\lambda)R_{12}(\lambda - \lambda'), \tag{A.1}$$

representing an operatorial identity on the tensor product of three vector spaces of the form $\mathcal{V}_{1/2} \otimes \mathcal{V}_{1/2} \otimes \mathcal{V}_S$. We have employed the standard embedding prescription in which the subscript indices specify on which spaces the operators act non-identically.

Lax matrices act on $\mathbb{C}^2 \otimes \mathcal{V}_S$, where $\mathcal{V}_S$ pertains to a unitary irreducible representation of 'Sklynanin quantum spins' $\hat{\mathcal{S}}^a$ (with $a \in \{0, 1, 2, 3\}$) of dimension $2S + 1$. The Lax operator

assumes the form

$$L(u) = \mathbb{1} \otimes \hat{\mathcal{S}}^0 + \sum_{a=1}^{3} W_a(\lambda)\, \sigma^a \otimes \hat{\mathcal{S}}^a. \tag{A.2}$$

**Quantum $R$-matrix.**   The $R$-matrix acts on two copies of $\mathcal{V}_{1/2} \cong \mathbb{C}^2$ which are regarded as *auxiliary* fundamental spins (subsequently represented by Pauli matrices $\sigma^a$). The $R$-matrix swaps (intertwines) the spectral parameters of the Lax operators.

The $R$-matrix has the 6-vertex form [112]

$$R(\lambda) = \mathbb{1} + \sum_{a=1}^{3} W_a(\lambda)\, \sigma^a \otimes \sigma^a, \tag{A.3}$$

with trigonometric amplitudes (Boltzmann weights)

$$W_1(\lambda) = W_2(\lambda) = \frac{\sinh(\eta)}{\sinh(\lambda+\eta)}, \qquad W_3(\lambda) = \frac{\tanh(\eta)}{\tanh(\lambda+\eta)}, \tag{A.4}$$

where

$$q = e^{\eta} \in \mathbb{R}_+ \tag{A.5}$$

is the quantum deformation parameter. The 6-vertex quantum $R$-matrix, cf. Eq. (A.3), is a specialization of more general 8-vertex elliptic $R$-matrix introduced by Sklyanin [72], obtained by sending the elliptic modulus to zero.

When embedded in $\mathcal{V}_{1/2} \otimes \mathcal{V}_{1/2} \otimes \mathcal{V}_{1/2}$, the $R$-matrix satisfies the quantum Yang–Baxter relation [77, 112]

$$R_{12}(\lambda-\lambda')R_{13}(\lambda)R_{23}(\lambda') = R_{23}(\lambda')R_{13}(\lambda)R_{12}(\lambda-\lambda'). \tag{A.6}$$

**Quantum Sklyanin algebra.**   The fundamental commutation relation (A.1) is equivalent to the Sklyanin quadratic algebra

$$[\hat{\mathcal{S}}^0, \hat{\mathcal{S}}^a] = \mathrm{i}\,\mathbb{J}_{bc}[\hat{\mathcal{S}}^b, \hat{\mathcal{S}}^c]_+, \qquad [\hat{\mathcal{S}}^a, \hat{\mathcal{S}}^b] = \mathrm{i}[\hat{\mathcal{S}}^0, \hat{\mathcal{S}}^c]_+, \tag{A.7}$$

which, in the general elliptic case, involves four generators $\hat{\mathcal{S}}^a$, with $a \in \{1,2,3,4\}$, and three 'structure constants' $\mathbb{J}_{ab} \equiv -(\mathbb{J}_a - \mathbb{J}_b)/\mathbb{J}_c$ (with indices $a, b, c$ mutually distinct) obeying the constraint $\mathbb{J}_{12} + \mathbb{J}_{23} + \mathbb{J}_{31} + \mathbb{J}_{12}\mathbb{J}_{23}\mathbb{J}_{31} = 0$. The Boltzmann weights specify an algebraic curve $W_a^2 - W_b^2 = \mathbb{J}_{ab}(W_c^2 - 1)$. The elliptic Sklyanin algebra admits two independent quadratic Casimir operators

$$\hat{\mathscr{C}}_0 = \sum_{a=0}^{3} (\hat{\mathcal{S}}^a)^2, \qquad \hat{\mathscr{C}}_1 = \sum_{a=1}^{3} \mathbb{J}_a (\hat{\mathcal{S}}^a)^2. \tag{A.8}$$

We are interested specifically in the trigonometric limit $\mathbb{J}_{12} \to 0$, where the Sklyanin algebra reduces to the quantum algebra $\mathcal{U}_q(\mathfrak{su}_2)$, representing a $q$-deformed quantum spin. In this limit, one finds $\hat{\mathscr{C}}_0 - \hat{\mathscr{C}}_1 = 1$, which leaves us with a single non-trivial Casimir operator of the form

$$\hat{\mathcal{C}}_q \equiv q^{-1}\hat{\mathcal{K}}^2 + q\hat{\mathcal{K}}^{-2} + (q - q^{-1})\hat{\mathcal{S}}^+\hat{\mathcal{S}}^-. \tag{A.9}$$

Moreover, writing $\kappa \equiv \sqrt{\mathbb{J}_{23}} = \sqrt{(\mathbb{J}_3 - \mathbb{J}_2)/\mathbb{J}_1}$ and introducing

$$\hat{\mathcal{S}}^{\pm} \equiv \hat{\mathcal{S}}^1 \pm \mathrm{i}\hat{\mathcal{S}}^2, \qquad \hat{\mathcal{K}} \equiv \hat{\mathcal{S}}^0 + \kappa\,\hat{\mathcal{S}}^3 = q^{\hat{\mathcal{S}}^3}, \tag{A.10}$$

the defining commutation relations of the trigonometric Sklyanin algebra read

$$[\hat{\mathcal{K}}, \hat{\mathcal{S}}^{\pm}] = \pm\kappa[\hat{\mathcal{K}}, \hat{\mathcal{S}}^{\pm}]_+, \qquad [\hat{\mathcal{S}}^+, \hat{\mathcal{S}}^-] = \frac{1}{\kappa}(\hat{\mathcal{K}}^2 - \hat{\mathcal{K}}^{-2}). \tag{A.11}$$

One can readily verify that these are indeed equivalent to standard $\mathcal{U}_q(\mathfrak{su}_2)$ algebraic relations upon introducing the $q$-deformation parameter $q = (1+\kappa)/(1-\kappa)$ and rescaling the generators,

$$\mathscr{S}^\pm \equiv \tfrac{1}{2}\sqrt{(1+\kappa)(1-\kappa)}\,\hat{\mathcal{S}}^\pm, \qquad \mathscr{K} \equiv \hat{\mathcal{K}}, \tag{A.12}$$

yielding

$$\mathscr{K}\,\mathscr{S}^\pm = q^{\pm 1}\,\mathscr{S}^\pm\,\mathscr{K}, \qquad [\mathscr{S}^+, \mathscr{S}^-] = \frac{\mathscr{K}^2 - \mathscr{K}^{-2}}{q - q^{-1}}. \tag{A.13}$$

## A.1 Semiclassical limit

We now perform the semiclassical limit. This is achieved, at the algebraic level, by expanding the fundamental commutation relation about the 'classical point' $q = 1$. Following Refs. [69, 113], we parameterize $q = e^\eta$, put $\eta = \hbar\varrho$ and subsequently expand the quantum $R$-matrix to the leading order in $\hbar$,

$$R(\lambda) = \mathbb{1} - 2\mathrm{i}\hbar\,r(\lambda) + \mathcal{O}(\hbar^2), \tag{A.14}$$

where [10]

$$r(\lambda) = -\frac{1}{2}\sum_{a=1}^{3} w_a(\lambda)\,\sigma^a \otimes \sigma^a, \tag{A.15}$$

is known as the classical $r$-matrix [69, 74, 113]. The classical amplitudes (Boltzmann weights) $w_a(\lambda)$ are similarly found by expanding the $W$-functions $W_a(\lambda) = \mathrm{i}\hbar\,w_a(\lambda) + \mathcal{O}(\hbar^2)$, and structure constatnts $\mathbb{J}_a = 1 - \hbar J_a + \mathcal{O}(\hbar^2)$, yielding

$$w_{1,2}(\lambda) = \frac{\varrho}{\sin(\lambda)}, \qquad w_3(\lambda) = \frac{\varrho}{\tan(\lambda)}, \tag{A.16}$$

with anisotropy parameter

$$\varrho \equiv \sqrt{J_3 - J_1} \in \mathbb{R}_+. \tag{A.17}$$

The classical structure constants now obey $J_{12} + J_{23} + J_{31} = 0$, and prescribe an algebraic curve $w_a^2(\lambda) - w_b^2(\lambda) = J_b - J_a$.

The classical $r$-matrix satisfies the classical analogue of the Yang–Baxter relation [69, 74]

$$[r_{12}(\lambda - \lambda'), r_{13}(\lambda)] + [r_{12}(\lambda - \lambda'), r_{23}(\lambda')] + [r_{13}(\lambda), r_{23}(\lambda')] = 0, \tag{A.18}$$

which can be readily deduced by expanding the Yang–Baxter relation (A.6) to the leading order $\mathcal{O}(\hbar^2)$.

**Sklyanin quadratic Poisson algebra.** In order to deduce the semiclassical limit of the fundamental commutation relation, one has to send $\hbar \to 0$ while keeping $\varrho$ and $\hbar S$ constant, thereby fixing the size of classical variables. This is achieved as follows. By expanding Eq. (A.1) to the leading order $\mathcal{O}(\hbar)$ and subsequently replacing commutators with Poisson brackets by virtue of the canonical correspondence principle [11]

$$[\circ, \circ] \to 2\mathrm{i}\hbar\{\circ, \circ\}, \tag{A.19}$$

the fundamental commutation relation reduces to the Sklyanin quadratic bracket [69]

$$\{L(\lambda)\,\substack{\otimes\\,}\,L(\lambda')\} = [r(\lambda - \lambda'), L(\lambda) \otimes L(\lambda')], \tag{A.20}$$

---

[10]There is freedom to choose the overall scale of the $r$-matrix. Here we adopt the common convention [60], which amounts to fixing a particular normalization of the Sklyanin Poisson algebra, cf. Eq. (A.23). Notice that our normalization differs from that of Ref. [69].

[11]Normalization of the Poisson bracket is set by the choice of the classical $r$-matrix, cf. Eq. (A.15).

with the classical Lax matrix [69]

$$L(\lambda) = \mathbb{1}\, \mathcal{S}^0 + \frac{1}{i} \sum_a w_a(\lambda)\sigma^a \mathcal{S}^a, \tag{A.21}$$

where we have simultaneously introduced classical 'Sklyanin variables' [12]

$$\hat{\mathcal{S}}^0 \to \hbar^{-1}\mathcal{S}^0, \qquad \hat{\mathcal{S}}^a \to \hbar^{-1}\varrho^{-1}\mathcal{S}^a, \quad \text{for} \quad a \in \{1,2,3\}. \tag{A.22}$$

Sklyanin quadratic bracket (A.20) provides a compact algebraic representation for the following *quadratic* Poisson algebra [69]

$$\{\mathcal{S}^0, \mathcal{S}^3\} = 0, \qquad \{\mathcal{S}^0, \mathcal{S}^a\} = \varrho^{-1}\epsilon_{abc}J_{bc}\mathcal{S}^b\mathcal{S}^c, \qquad \{\mathcal{S}^a, \mathcal{S}^b\} = -\varrho\,\epsilon_{abc}\mathcal{S}^0\mathcal{S}^c, \tag{A.23}$$

with classical structure constants $J_{ab} \equiv J_a - J_b$. Poisson algebra (A.23) admits two independent Casimir functions

$$\mathcal{C}_0 \equiv \sum_{a=1}^3 \left(\mathcal{S}^a\right)^2, \qquad \mathcal{C}_1 \equiv \left(\mathcal{S}^0\right)^2 - \left(\mathcal{S}^3\right)^2. \tag{A.24}$$

By prescribing their values $c_0$ and $c_1$, respectively, variables $\mathcal{S}^a$ takes values on a two-dimensional non-degenerate Poisson submanifold. By fixing them to

$$c_0 = \sinh^2(\varrho), \qquad c_1 = 1, \tag{A.25}$$

the manifold is topologically equivalent to a two-sphere; $\{\mathcal{S}^a\}_{a=1}^3$ lie on a two-sphere, whereas $\mathcal{S}^0 > 0$ is uniquely determined by $\mathcal{S}^3$ through $\mathcal{C}_1$.

Introducing linear combinations $\mathcal{K}^\pm \equiv \mathcal{S}^0 \pm \mathcal{S}^3$ and putting $\mathcal{K} \equiv \mathcal{K}^+$, we have $\mathcal{K}^+\mathcal{K}^- = \mathcal{C}_1$, yielding [114]

$$\mathcal{A}_q: \qquad \{\mathcal{K}, \mathcal{S}^\pm\} = \mp i\varrho\, \mathcal{S}^\pm\mathcal{K}, \qquad \{\mathcal{S}^+, \mathcal{S}^-\} = -\frac{i\varrho}{2}(\mathcal{K}^2 - \mathcal{K}^{-2}). \tag{A.26}$$

The above Poisson relation also follow from quantum algebraic relations (A.11) upon rescaling $\kappa$ as $\kappa = \hbar\varrho$, applying the canonical corespondence (A.19), and finally sending $\hbar \to 0$. Poisson algebra $\mathcal{A}_q$, with a pair of Casimir functions $\mathcal{C}_0, \mathcal{C}_1$, can be understood as the classical counterpart of $\mathcal{U}_q(\mathfrak{su}_2)$.

## B Factorization in Weyl variables

We provide explicit prescriptions for the elementary exchanges of Weyl matrix $U$ and $V$ that appear in the decomposition of the Lax matrix (cf. Section 2), written in the form of symplectic transformations on Weyl pairs $(x_\ell, y_\ell)$:

(i) $U_1 - U_2$ exchange (35)

$$x_1' = x_1, \tag{B.1}$$

$$x_2' = x_2 + (u_1 - u_2)\frac{x_1 x_2 v_2(u_1 + u_2 y_2^2) - x_2^2(1 + u_1 u_2 v_2^2 y_2^2)}{x_1 u_1 u_2 v_2(1 - y_2^2) + u_2 x_2(u_1^2 v_2^2 y_2^2 - 1)}, \tag{B.2}$$

$$(y_1')^2 = y_1^2 - (u_1^2 - u_2^2)\frac{v_2 x_1 y_1^2(y_2^2 - 1)}{u_1 x_2(u_2^2 v_2^2 y_2^2 - 1) + x_1 v_2 u_2^2(1 - y_2^2)}, \tag{B.3}$$

$$(y_2')^2 = y_2^2 + (u_1^2 - u_2^2)\frac{v_2 x_1 y_2^2(y_2^2 - 1)}{u_1 x_2(u_2^2 v_2^2 y_2^2 - 1) + x_1 v_2 u_1^2(1 - y_2^2)}, \tag{B.4}$$

---

[12]Our normalization here slightly differs from the one in Ref. [60] in that $\mathcal{S}^a$ for $a \in \{1,2,3\}$ were rescaled by a factor of $\varrho^{-1}$.

and in 'Trotter parametrization' (41) [13]

$$x'_1 = x_1, \tag{B.5}$$

$$x'_2 = x_2 \frac{x_1(y_2 w^{-1} - w y_2^{-1}) + x_2(v^{-2} y_2^{-1} - y_2 v^2)}{x_1(y_2 - y_2^{-1}) + x_2(w^{-1} v^{-2} y_2^{-1} - y_2 w v^2)}, \tag{B.6}$$

$$(y'_1)^2 = y_1^2 \frac{x_1(y_2 - y_2^{-1})w + x_2(y_2^{-1} v^{-2} - y_2 v^2)}{x_1(y_2 - y_2^{-1})w^{-1} + x_2(y_2^{-1} v^{-2} - y_2 v^2)}, \tag{B.7}$$

$$(y'_2)^2 = y_2^2 \frac{x_1(y_2 - y_2^{-1})w^{-1} + x_2(y_2^{-1} v^{-2} - y_2 v^2)}{x_1(y_2 - y_2^{-1})w + x_2(y_2^{-1} v^{-2} - y_2 v^2)}, \tag{B.8}$$

(ii) $V_1 - V_2$ exchange (37)

$$x'_1 = x_1 - (v_1 - v_2)\frac{x_1 x_2 u_1(v_2 + v_1 y_1^2) - x_1^2(1 + v_1 v_2 u_1^2 y_1^2)}{x_2 v_1 v_2 u_1(1 - y_1^2) + v_1 x_1(u_1^2 v_2^2 y_1^2 - 1)}, \tag{B.9}$$

$$x'_2 = x_2, \tag{B.10}$$

$$(y'_1)^2 = y_1^2 - (v_1^2 - v_2^2)\frac{u_1 x_2 y_1^2(y_1^2 - 1)}{v_2 x_1(u_1^2 v_1^2 y_1^2 - 1) + x_2 u_1 v_2^2(1 - y_1^2)}, \tag{B.11}$$

$$(y'_2)^2 = y_2^2 + (v_1^2 - v_2^2)\frac{u_1 x_2 y_2^2(y_1^2 - 1)}{v_2 x_1(u_1^2 v_1^2 y_1^2 - 1) + x_2 u_1 v_1^2(1 - y_1^2)}, \tag{B.12}$$

and in 'Trotter parametrization' (41)

$$x'_1 = x_1 \frac{x_1(y_1^{-1} v^{-2} - y_1 v^2) + x_2(y_1 w^{-1} - w y_1^{-1})}{x_1(y_1^{-1} w^{-1} v^{-2} - y_1 w v^2) + x_2(y_1 - y_1^{-1})}, \tag{B.13}$$

$$x'_2 = x_2, \tag{B.14}$$

$$(y'_1)^2 = y_1^2 \frac{x_1(y_1^{-1} v^{-2} - v^2 y_1) + x_2 w^{-1}(y_1 - y_1^{-1})}{x_1(y_1^{-1} v^{-2} - v^2 y_1) + x_2 w(y_1 - y_1^{-1})}, \tag{B.15}$$

$$(y'_2)^2 = y_2^2 \frac{x_1(y_1^{-1} v^{-2} - v^2 y_1) + x_2 w(y_1 - y_1^{-1})}{x_1(y_1^{-1} v^{-2} - v^2 y_1) + x_2 w^{-1}(y_1 - y_1^{-1})}. \tag{B.16}$$

The combined $U - V$ exchange (in Trotter parametrization (41)) gives the two-body map $\Phi$

$$x'_1 = x_1 \frac{x_1 Y_1(v,0) - x_2 Y_1(1,-\tau)}{x_1 Y_1(v,\tau) - x_2 Y_1(1,0)}, \qquad x'_2 = x_2 \frac{x_1 Y_2(1,-\tau) - x_2 Y_2(v,0)}{x_1 Y_2(1,0) - x_2 Y_2(v,\tau)}, \tag{B.17}$$

$$y'^2_1 = y_1^2 \frac{f_2^-}{f_1^+}, \qquad\qquad\qquad y'^2_2 = y_2^2 \frac{f_1^+}{f_2^-}, \tag{B.18}$$

with

$$f_j^{\pm} = x_{j\pm1}^2 w Y_j(1,0) Y_{j\pm1}(v,0) + x_j^2 w^{-1} Y_{j\pm1}(1,0) Y_j(v,0)$$
$$- x_{j\pm1} x_j \left( Y_{j\pm1}(v,0) Y_j(v,0) + Y_{j\pm1}(1,0) Y_j(1,0) - y_j y_{j\pm1}^{-1}(w - w^{-1})^2 \right), \tag{B.19}$$

$$Y_j(v,\tau) = v^2 y_j w - v^{-2} y_j^{-1} w^{-1}. \tag{B.20}$$

[13]Where here and in the following we use $w = e^{i\varrho\tau}$.

# C  Isospectral flows and local conservation laws

A key implication of the discrete zero-curvature condition (3) is the existence of infinitely many conserved quantities which are in mutual involution. As we shortly demonstrate, these are a corollary of isospectrality of the transfer function. We accordingly introduce a staggared monodromy matrix $T_\tau(\lambda) \equiv T_\tau(\{S_\ell\}; \lambda)$, defined as a right-to-left ordered product of Lax matrices along a zig-zag path (i.e. initial sawtooth) of length $L$

$$T_\tau(\lambda) = L(S_L; \lambda^+) L(S_{L-1}; \lambda^-) \cdots L(S_2; \lambda^+) L(S_1; \lambda^-), \tag{C.1}$$

where $\lambda^\pm \equiv \lambda \pm \tau/2$. Here and subsequently we leave the dependence on anisotropy $\varrho$ implicit. By tracing out the auxiliary $\mathbb{C}^2$ space, we obtain the transfer function

$$t_\tau(\lambda) = \mathrm{Tr}\, T_\tau(\lambda). \tag{C.2}$$

**Isospectral evolution.**  By virtue of the zero-curvature condition (3) and cyclic property of the trace, it follows that

$$t_\tau(\lambda) \circ \Phi_\tau^{\mathrm{even}} = t_{-\tau}(\lambda), \qquad t_{-\tau}(\lambda) \circ \Phi_\tau^{\mathrm{odd}} = t_\tau(\lambda), \tag{C.3}$$

which in combination imply invariance under the time propagation (6)

$$t_\tau(\lambda) \circ \Phi_\tau^{\mathrm{full}} = t_\tau(\lambda). \tag{C.4}$$

The quadratic bracket (A.20) can be immediately lifted onto the level of monodromies

$$\left\{ T_\tau(\lambda) \overset{\otimes}{,} T_\tau(\mu) \right\} = \left[ r(\lambda - \mu), T_\tau(\lambda) \otimes T_\tau(\mu) \right]. \tag{C.5}$$

By subsequently taking the trace over $\mathcal{C}^2$, one readily finds that transfer functions Poisson commute,

$$\left\{ t_\tau(\lambda), t_\tau(\lambda') \right\} = 0, \tag{C.6}$$

for any pair of complex spectral parameters $\lambda, \lambda' \in \mathbb{C}$ and fixed $\tau$. Preservation of the characteristic polynomial of $T_\tau(\lambda)$ is referred to as isospectrality.

**Local conservation laws.**  Because the transfer function $t_\tau(\lambda)$ depends analytically on $\lambda$, it can be used as a generating function of integrals of motion. In the thermodynamic limit, $L \to \infty$, this provides infinitely many functionally independent conserved quantities. Expanding $t_\tau(\lambda)$ in a power series in $\lambda$ nonetheless produces complex-valued and *non-local* conserved quantities. In order to construct real conservation laws, it is useful to utilize the following involutions of the Lax matrix

$$\overline{L(\lambda)} = \sigma^2 L(\bar{\lambda}) \sigma^2, \qquad L^T(\lambda) = \sigma^2 L(-\lambda) \sigma^2, \tag{C.7}$$

which likewise apply to the monodromy matrix $T_\tau(\lambda)$. It follows that the moduli of the traces are also in mutual involution, and their expansion in $\lambda$ thus yields manifestly real conserved quantities.

One can in fact extract two seperate infinite towers of *local* conserved quantities, denoting them by $Q_{k;\tau}^\pm$, using the standard approach (see, for example, Ref. [60]) by expanding the logarithmic derivatives of the modulus of $t_\tau(\lambda)$ around either of the two distinguished points $\lambda_0^\pm = \mathrm{i} \mp \frac{\tau}{2}$ at which the Lax matrix degenerates (becoming a projector of rank one),

$$\mathrm{Det}\, L(\lambda_0^\pm) = 0, \qquad L(\lambda_0^\pm) = |\alpha_\pm\rangle \langle \beta_\pm|, \tag{C.8}$$

yielding two independent sequences of local charges

$$Q_{k;\tau}^{\pm} = \frac{\partial^k}{\partial \lambda^k} \log |t_\tau(\lambda)|^2 \bigg|_{\lambda=\lambda_0^{\pm}}. \tag{C.9}$$

Taking the continuous time limit $\tau \to 0$, both branches merge into the local conservation laws of the anisotropic lattice Landau-Lifshitz model.

**Parity symmetry.** The transfer function $t_\tau(\lambda)$ is $\mathbb{Z}_2$-invariant under the spin-reversal transformation ($\pi$-rotation of the canonical spin **S** around the $x$-axis)

$$\mathcal{F}_1: \qquad S^3 \mapsto -S^3, \qquad S^{\pm} \mapsto S^{\mp}, \tag{C.10}$$

implying $\mathcal{F}_1(\mathcal{K}) = \mathcal{K}^{-1}$. At the level of the Lax matrix, $\mathcal{F}_1$ represents conjugation by Pauli matrix $\sigma^1$,

$$\mathcal{F}_1: \qquad L(\lambda) \mapsto \sigma^1 L(\lambda)\sigma^1, \qquad T(\lambda) \mapsto \sigma^1 T(\lambda)\sigma^1, \tag{C.11}$$

which implies invariance $\mathcal{F}_1(t_\tau(\lambda)) = t_\tau(\lambda)$ and hence charges $Q_{k;\tau}^{\pm}$ are all invariant under $\mathcal{F}_1$.

# D Reductions

The anisotropic model of interacting spin in discrete space-time, introduced in Section 2, admits several important reductions. There are three distinct parameters that can be taken to zero: anisotropy $\varrho$, discrete time-step $\tau$, and lattice spacing $\Delta$ (which we have kept fixed $\Delta = 1$ throughout the text). The full set of models which descend from $\Phi_{\tau,\varrho}$ can be therefore arranged on the vertices of a cube, as depicted in Figure 10. We briefly describe them below. [14]

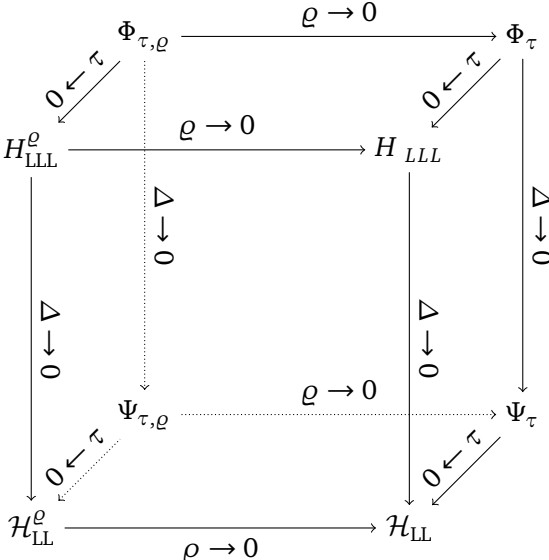

Figure 10: Reductions of the anisotropic discrete-time propagator $\Phi_{\tau,\varrho}$ (upper-left corner), obtained via taking various combinations of $\varrho, \Delta, \tau \to 0$ limits. The complete reduction obtained by taking all three limit yields the isotropic Landau-Lifshitz field theory $\mathcal{H}_{\mathrm{LL}}$ (lower-right corner).

---

[14] We shall omit discussing $\Psi_\tau^\varrho$ and $\Psi_\tau$, which correspond to (anisotropic) 'kicked' field theories in discrete time.

$\Phi_\tau$ **– isotropic discrete space-time model.** By letting the anisotropy parameter to zero, $\varrho \to 0$, the anisotropy Lax matrix given by Eq. (15) becomes the Lax matrix of the isotropic lattice Landau–Lifshitz model [69, 115]

$$L^{(0)}(\lambda, \mathbf{S}) = \mathbb{1} + \frac{1}{\mathrm{i}\lambda}\mathbf{S}\cdot\boldsymbol{\sigma}, \tag{D.1}$$

where $\boldsymbol{\sigma} = (\sigma^1, \sigma^2, \sigma^3)$ is a vector of Pauli matrices. The corresponding discrete zero-curvature condition around a plaquette,

$$L_2^{(0)}(\lambda + \tau/2)L_1^{(0)}(\lambda - \tau/2) = L_2'^{(0)}(\lambda - \tau/2)L_1'^{(0)}(\lambda + \tau/2), \tag{D.2}$$

generates a two-body symplectic propagator [55]

$$\Phi_\tau(\mathbf{S}_1, \mathbf{S}_2) = \frac{1}{s^2 + \tau^2}\left(s^2\mathbf{S}_1 + \tau^2\mathbf{S}_2 + \tau\mathbf{S}_1 \times \mathbf{S}_2, s^2\mathbf{S}_2 + \tau^2\mathbf{S}_1 + \tau\mathbf{S}_2 \times \mathbf{S}_1\right), \tag{D.3}$$

$$s^2 = \frac{1}{2}\left(1 + \mathbf{S}_1 \cdot \mathbf{S}_2\right). \tag{D.4}$$

$H_{\mathrm{LLL}}^\varrho$ **– uniaxially anisotropic lattice Landau-Lifshitz model.** By sending $\tau \to 0$, the discrete-time propagator becomes the equation of motion in continuous time,

$$\frac{\mathrm{d}}{\mathrm{d}t}\mathcal{S}_\ell^a = \left\{\mathcal{S}^a, H_{\mathrm{LLL}}^\varrho\right\}, \tag{D.5}$$

generated by a Hamiltonian

$$H_{\mathrm{LLL}}^\varrho \simeq \sum_{\ell=1}^{L}\log\left[\sinh^2\varrho\,\mathcal{S}_\ell^0\mathcal{S}_{\ell+1}^0 + \mathcal{S}_\ell^1\mathcal{S}_{\ell+1}^1 + \mathcal{S}_\ell^2\mathcal{S}_{\ell+1}^2 + \cosh^2\varrho\,\mathcal{S}_\ell^3\mathcal{S}_{\ell+1}^3\right]. \tag{D.6}$$

$H_{\mathrm{LLL}}$ **– isotropic lattice Landau-Lifshitz model.** Starting either from Eq. (D.4) and sending $\tau \to 0$, or from Eq. (D.5) and sending $\varrho \to 0$, one ends up with the Hamiltonian dynamics of the isotropic lattice Landau-Lifshitz model. In terms of canonical spins $\mathbf{S}$, the equations of motion read [115]

$$\frac{\mathrm{d}}{\mathrm{d}t}\mathbf{S}_\ell = \{\mathbf{S}_\ell, H_{\mathrm{LLL}}\} = \frac{\mathbf{S}_\ell \times \mathbf{S}_{\ell-1}}{1 + \mathbf{S}_\ell \cdot \mathbf{S}_{\ell-1}} + \frac{\mathbf{S}_\ell \times \mathbf{S}_{\ell+1}}{1 + \mathbf{S}_\ell \cdot \mathbf{S}_{\ell+1}}, \tag{D.7}$$

with a Hamiltonian of the form $H_{\mathrm{LLL}} = \sum_{\ell=1}^{L}\log\left(1 + \mathbf{S}_\ell \cdot \mathbf{S}_{\ell+1}\right)$.

$\mathcal{H}_{\mathrm{LL}}^\varrho$ **– anisotropic Landau–Lifshitz field theory.** To obtain the full continuum limit, we reinstate the lattice spacing $\Delta$ and subsequently retain smooth lattice spin configurations $\mathbf{S}_\ell = \mathbf{S}(x = \Delta\ell)$ by keeping only variations at the leading non-trivial order $\mathcal{O}(\Delta^2)$:

$$\mathbf{S}_{\ell\pm1} = \mathbf{S}(x) \pm \Delta\partial_x\mathbf{S} + \frac{1}{2}\Delta^2\,\partial_x^2\mathbf{S} + \mathcal{O}(\Delta^3). \tag{D.8}$$

Upon taking the continuum limit $\Delta \to 0$, both the Sklyanin variables and the anisotropy parameter have to be accordingly rescaled by the lattice spacing, namely

$$\mathcal{S}^0 = 1 + \mathcal{O}(\Delta^2), \qquad \mathcal{S}^j = \Delta\mathbf{S}^a + \mathcal{O}(\Delta^2) \quad \text{for} \quad a = 1, 2, 3, \tag{D.9}$$

and $\varrho \to \Delta\varrho$. This way, one recovers the anisotropic Landau-Lifshitz field theory [59, 60]

$$\mathcal{H}_{\mathrm{LL}}^\varrho = \frac{1}{2}\int_0^L \mathrm{d}x\,\left[(\partial_x\mathbf{S})^2 - \mathbf{S}\cdot\mathcal{J}\mathbf{S}\right], \qquad \mathcal{J} = \mathrm{diag}(J_1, J_2, J_3), \tag{D.10}$$

(where $J_1 = J_2$ and $J_3 - J_1 = \varrho^2$) with the equation of motion [59, 60]

$$\partial_t\mathbf{S}(x) = \{\mathbf{S}(x), \mathcal{H}_{\mathrm{LL}}^\varrho\} = \mathbf{S} \times \partial_x^2\mathbf{S} + \mathbf{S} \times \mathcal{J}\mathbf{S}. \tag{D.11}$$

$\mathcal{H}_{\mathrm{LL}}$ **– isotropic Landau-Lifshitz field theory.** By finally sending $\varrho \to 0$, Eq. (D.10) becomes the isotropic Landau-Lifshitz field theory [58, 60]

$$\mathcal{H}_{\mathrm{LL}} = \frac{1}{2} \int_0^L \mathrm{d}x \, (\partial_x \mathbf{S})^2, \qquad \partial_t \mathbf{S} = \{\mathbf{S}, \mathcal{H}_{\mathrm{LL}}\} = \mathbf{S} \times \partial_x^2 \mathbf{S}. \tag{D.12}$$

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
