# Peer review of "Anisotropic Landau-Lifshitz Model in Discrete Space-Time"

_SciPost Physics, doi:SciPost Phys. 11, 051 (2021)_

## Round 1 · Referee Report · Anonymous (Referee 1) · 2021-7-18

Report

The authors construct an integrable space-time discretization of the classical anisotropic Landau-Lifshitz model. The paper is timely, interesting and well-written. I have only a couple of scientific questions/comments:

1) On p6-8 the authors work with “Sklyanin variables” rather than canonical spin variables, which they justify on the grounds that it is “to facilitate the computation”. However, the space-time continuum limits of the models in question are usually discussed in terms of standard spin variables. Also, the explicit formulas for the propagator in Appendix B seem much more complicated than the expressions that arise at the isotropic point in terms of standard spin variables Eqs. D3 and D4. Could the authors clarify why Sklyanin variables are more natural here, and what advantage they offer compared to ordinary spins?

2) On p.18-19 the authors conjecture that (quasi)-local charges protect the easy-plane spin current, as for the spin-1/2 quantum XXZ model. They also note that for their model, the standard local charges arising from the transfer matrix are spin-flip even. However, this is also true in the quantum case. Could the authors elaborate on the obstacles to writing down classical Z-charges? These have been conjectured to arise as a semiclassical limit of quantum Z-charges for some time (e.g. Ref 71).

Finally, I noticed the following typos:
p2: Can be both -> can both be,
p2, p13,p29: Landau-Lifhsitz -> Landau-Lifshitz
Fig. 2 caption: comprising of -> comprising
p6: Algebra A_q becomes non-degenerate-> The algebra A_q…
p8: To enforce it, variable x -> …the variable x
p8: formuale -> formulae, p20: formuae -> formulae
p10: simplifies considerably -> simplify considerably
p11: There is a number -> there are a number
p15: converging -> converges
p15,16,17,20: consistently - > consistent
p20: corresponding the easy-axis -> corresponding to the…
  • validity: -
  • significance: -
  • originality: -
  • clarity: -
  • formatting: -
  • grammar: -

Author:  Žiga Krajnik  on 2021-07-27  [id 1618]

(in reply to Report 1 on 2021-07-18)
Category:
remark
answer to question

We thank the referee for his careful reading of the manuscript and the positive appraisal of our work.
The answers to the referee's questions follow below:

(1) One can quickly recognize that, upon deforming the model, canonical spins are no longer the most natural variables to work with. We remind the reader that in the quantum setting, the Sklyanin algebra specifies the commutation relations for an elliptic quantum group, reducing to the widely known q-deformed su(2) algebra in the trigonometric limit. In the paper, we deal with Sklynain's Poisson algebra, which is its classical counterpart. The classical Lax matrix (and likewise the Hamiltonian) can be elegantly written in terms of Sklyanin variables (see also the textbook [61]), and this further facilitates its decomposition in terms of Weyl variables. The most economic explicit parametrization of the two-body map we could achieve was in terms of Sklyanin variables. We could not recognize any advantage of working with canonical spins. Those are only used at the very end in applications, given that we wish to interpret our model as discrete dynamics of magnetization.

(2) To avoid running to excessive length we had to come to a stop and not pursue this matter further. We note that the whole discussion regarding the existence and physical significance of "classical Z-charges" has been given before, e.g. in Ref.[71] (as also noted by the referee) which invokes the Mazur-Suzuki bound and invariance under the spin-reversal transformation. Moreover in the present paper we do not imply that there is a fundamental obstacle to constructing such charges. We have made substantial progress on this question after the completion of the study. There are nonetheless certain delicate technical aspects that deserve a careful and focused discussion which we prefer to report elsewhere. We hope to have an update soon.

---

## Round 2 · Referee Report · Anonymous (Referee 2) · 2021-8-16

Report
The authors construct an integrable model of classical spins that can be viewed as a discrete time version of the anisotropic Landau-Lifshitz ferromagnet. This generalizes recent work by two of the authors on the construction of an SU(2) symmetric ferromagnet in Ref. [56].
The authors' construction is based on a discrete zero-curvature condition on an auxiliary light-cone lattice, the solution of which defines the elementary "propagator" that maps pairs of physical spins forward in time. Arguably the key result of the work is the explicit construction of this propagator by exploiting certain factorization properties of the Lax operators that solve the discrete zero curvature conditions. This is a very interesting and to my knowledge original construction.
The second part of the work focuses on a numerical analysis of transport properties in the newly constructed anisotropic ferromagnet, in particular on magnetization transport in thermal equilibrium as
characterized by the spin Drude weight and spin diffusion constant. This analysis is timely and ties in very nicely with recent developments in the field such as cross-overs from ballistic to superdiffusive behaviors in the vicinity of the isotropic (SU(2) invariant) point. Moreover, the numerical results are suggestive of the existence of yet unknown quasi-local conservation laws in the model, and it clearly would be interesting to try to construct them explicitly.
The manuscript is well written and makes a number of new and interesting contributions to the field as detailed above. I recommend publication in its current form.
The authors' construction is based on a discrete zero-curvature condition on an auxiliary light-cone lattice, the solution of which defines the elementary "propagator" that maps pairs of physical spins forward in time. Arguably the key result of the work is the explicit construction of this propagator by exploiting certain factorization properties of the Lax operators that solve the discrete zero curvature conditions. This is a very interesting and to my knowledge original construction.
The second part of the work focuses on a numerical analysis of transport properties in the newly constructed anisotropic ferromagnet, in particular on magnetization transport in thermal equilibrium as
characterized by the spin Drude weight and spin diffusion constant. This analysis is timely and ties in very nicely with recent developments in the field such as cross-overs from ballistic to superdiffusive behaviors in the vicinity of the isotropic (SU(2) invariant) point. Moreover, the numerical results are suggestive of the existence of yet unknown quasi-local conservation laws in the model, and it clearly would be interesting to try to construct them explicitly.
The manuscript is well written and makes a number of new and interesting contributions to the field as detailed above. I recommend publication in its current form.

---

## Round 2 · Referee Report · Anonymous (Referee 2) · 2021-8-18

Strengths
- original construction of an integrable discretization of the anisotropic Landau-Lifzhitz ferromagnet
- well written manuscript
- numerical results timely with regards to recent works on cross-overs from ballistic to superdiffusive behaviors
Weaknesses
- none.
Report
The authors construct an integrable model of classical spins that can
be viewed as a discrete time version of the anisotropic
Landau-Lifshitz ferromagnet. This generalizes recent work by two of
the authors on the construction of an SU(2) symmetric ferromagnet in
Ref. [56].
The authors' construction is based on a discrete zero-curvature
condition on an auxiliary light-cone lattice, the solution of which
defines the elementary "propagator" that maps pairs of physical spins
forward in time. Arguably the key result of the work is the explicit
construction of this propagator by exploiting certain factorization
properties of the Lax operators that solve the discrete zero curvature
conditions. This is a very interesting to to my knowledge original
construction.
The second part of the work focuses on a numerical analysis of
transport properties in the newly constructed anisotropic ferromagnet,
in particular on magnetization transport in thermal equilibrium as
characterized by the spin Drude weight and spin diffusion
constant. This analysis is timely and ties in very nicely with recent
developments in the field such as cross-overs from ballistic
to superdiffusive behaviors in the vicinity of the isotropic (SU(2)
invariant) point. Moreover, the numerical results are suggestive of
the existence of yet unknown quasi-local conservation laws in the
model, and it clearly would be interesting to try to construct them
explicitly.
The manuscript is well written and makes a number of new and interesting
contributions to the field as detailed above. I recommend publication in its
current form.
be viewed as a discrete time version of the anisotropic
Landau-Lifshitz ferromagnet. This generalizes recent work by two of
the authors on the construction of an SU(2) symmetric ferromagnet in
Ref. [56].
The authors' construction is based on a discrete zero-curvature
condition on an auxiliary light-cone lattice, the solution of which
defines the elementary "propagator" that maps pairs of physical spins
forward in time. Arguably the key result of the work is the explicit
construction of this propagator by exploiting certain factorization
properties of the Lax operators that solve the discrete zero curvature
conditions. This is a very interesting to to my knowledge original
construction.
The second part of the work focuses on a numerical analysis of
transport properties in the newly constructed anisotropic ferromagnet,
in particular on magnetization transport in thermal equilibrium as
characterized by the spin Drude weight and spin diffusion
constant. This analysis is timely and ties in very nicely with recent
developments in the field such as cross-overs from ballistic
to superdiffusive behaviors in the vicinity of the isotropic (SU(2)
invariant) point. Moreover, the numerical results are suggestive of
the existence of yet unknown quasi-local conservation laws in the
model, and it clearly would be interesting to try to construct them
explicitly.
The manuscript is well written and makes a number of new and interesting
contributions to the field as detailed above. I recommend publication in its
current form.

---

## Round 2 · Author Response

Revised version.

---

## Round 2 · List of Changes

-We have corrected the typos and grammar mistakes pointed out by the referee and several others.
-We have added a few missing DOIs.
-We have added a few missing DOIs.

---

## Editorial Decision

published